# Dynamic Tuning Towards Parameter and Inference Efficiency for ViT Adaptation

**Wangbo Zhao**[1,2*]    **Jiasheng Tang**[2,3†]    **Yizeng Han**[2,4]    **Yibing Song**[2,3]    **Kai Wang**[1]

**Gao Huang**[4]    **Fan Wang**[2]    **Yang You**[1†]

[1]National University of Singapore    [2]DAMO Academy, Alibaba Group

[3]Hupan Laboratory    [4]Tsinghua University

Code: `https://github.com/NUS-HPC-AI-Lab/Dynamic-Tuning`

## Abstract

Existing parameter-efficient fine-tuning (PEFT) methods have achieved significant success on vision transformers (ViTs) adaptation by improving parameter efficiency. However, the exploration of enhancing inference efficiency during adaptation remains underexplored. This limits the broader application of pre-trained ViT models, especially when the model is computationally extensive. In this paper, we propose **Dy**namic **T**uning (DyT), a novel approach to improve both parameter and inference efficiency for ViT adaptation. Specifically, besides using the lightweight adapter modules, we propose a *token dispatcher* to distinguish informative tokens from less important ones, allowing the latter to *dynamically* skip the original block, thereby reducing the redundant computation during inference. Additionally, we explore multiple design variants to find the best practice of DyT. Finally, inspired by the mixture-of-experts (MoE) mechanism, we introduce an enhanced adapter to further boost the adaptation performance. We validate DyT across various tasks, including image/video recognition and semantic segmentation. For instance, DyT achieves superior performance compared to existing PEFT methods while evoking only 71% of their FLOPs on the VTAB-1K benchmark.

## 1   Introduction

With the remarkable success of Vision Transformers (ViTs) [20, 53, 26], fine-tuing pre-trained ViT on other data domains [90] or task applications [89, 36, 60, 79, 31] is becoming a common practice. However, as model sizes increase [88, 16, 52], the associated adaptation costs become prohibitive due to the burden of fine-tuning and inference on the target task. Parameter-efficient fine-tuning (PEFT) methods (*e.g.* AdaptFormer [12], LoRA [34], and VPT [36]), are proposed to tackle the tuning problem by reducing tunable model parameters. They usually update a small amount of parameters while keeping the original model fixed, which reduces learnable parameters effectively while maintaining fine-tuning accuracy.

Despite the extensive research in parameter efficiency, the *inference efficiency* on target tasks is less explored. We numerically demonstrate the inference efficiency of three representative PEFT methods in Figure. 1(a), revealing that none of them reduces the computation during inference compared to full tuning. This limitation poses challenges for adapting pre-trained ViT to downstream tasks, particularly when the model is computationally extensive. To this end, we aim to unify both parameter and inference perspectives for efficient ViT adaptations.

---

[*]Work done during an internship at DAMO Academy, Alibaba Group, wangbo.zhao96@gmail.com

[†]Corresponding authors, jiasheng.tjs@alibaba-inc.com, youy@comp.nus.edu.sg

38th Conference on Neural Information Processing Systems (NeurIPS 2024).

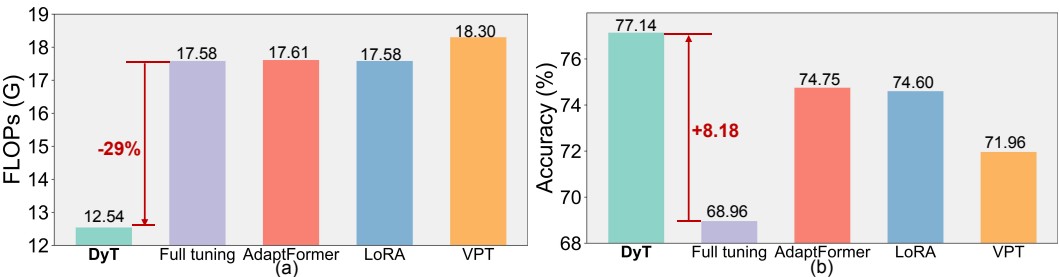

Figure 1: **FLOPs and Accuracy of ViT-B/16 [20] on VTAB-1K [89].** "Full tuning" denotes that all parameters are fine-tuned. AdaptFormer [12], LoRA [34] and VPT [36] are typical PEFT methods.

Dynamic networks [83, 67, 48] demonstrate that token pruning helps to reduce model inference costs. However, these designs primarily focus on pre-training from scratch or full tuning on the same dataset, without considering data domain transfer. Furthermore, the token pruning operation constraints these methods on visual recognition scenarios, limiting their further applications *e.g.* dense prediction [93]. Our motivation thus arises from how to develop dynamic modules together with PEFT methods to enhance both parameter and inference efficiency during ViT adaptation, which should also suit a wide range of visual tasks.

In this paper, we introduce **Dy**namic **T**uning (DyT), an efficient approach for ViTs adaptation that achieves both parameter and inference efficiency. Specifically, we design a token dispatcher for each transformer block learning to dynamically determine whether a token should be activated or deactivated. Only those activated tokens will traverse the entire block and an extra lightweight adapter module, while the remaining tokens skip the original block and will solely be processed by the adapter. DyT does not reduce the total number of tokens, making it suitable for both visual recognition and dense prediction scenarios. Additionally, since the impact of token skipping during adaptation has not been explored, we propose four model variants to determine the best practice for DyT. Finally, as the adapter module is set to process all the visual tokens, we propose a mixture-of-expert(MoE)-based adapter design that further enhances token processing with negligible additional FLOPs. Through these designs, our DyT fulfills both parameter and inference efficiency for ViTs adaptation.

Comprehensive evaluations for DyT are conducted from multiple perspectives. In the image domain, as shown in Figure 1(b), DyT surpasses existing PEFT methods while consuming only 71% of the ViT-B FLOPs on the VTAB-1K benchmark [89]. When visual tokens are scaled up from images to videos, our DyT shows superior generalization ability on action recognition benchmarks, *e.g.* K400 [10] and SSV2 [25], with a reduction of 37GFLOPs. In the scenario where labels are scaled up from recognition to segmentation, our DyT even outperforms full tuning on ADE20K [93] with 21GFLOPs reduction. These experimental results indicate that the proposed DyT is efficient in both parameter and inference across various visual tasks and directs a promising field for efficient model adaptation.

## 2 Related Works

**Parameter-efficient fine-tuning.** Parameter-efficient fine-tuning (PEFT) is designed to adapt a pre-trained model to downstream tasks by only tuning a small part of the parameters. Existing PEFT methods can broadly be categorized into three groups: adapter-based methods, re-parametrization-based methods, and prompt-based methods. Adapter-based methods [12, 33, 62, 38, 14] insert some tiny modules into the original model and only these inserted modules are updated during fine-tuning. Re-parametrization approaches [87, 34, 47, 21, 18] usually cooperate with reparameterization techniques, directly modifying specific parameters in the pre-trained model. Prompt-based methods [36, 2, 95, 94, 40, 9, 8] involve appending a small number of learnable tokens to input sequences, thereby adapting intermediate features in frozen layers. Please refer to [84, 46] to find more comprehensive reviews.

However, PEFT methods primarily focus on improving parameter efficiency during fine-tuning while overlooking inference cost reduction. In this paper, we propose to improve the parameter efficiency and address inference hosts simultaneously for efficient ViT adaptation.

**Dynamic neural networks.** Dynamic neural networks[28, 63, 92] can dynamically adjust their architectures based on the input data, which enables them to control the computational redundancy based on input data. Existing methods have explored dynamic layer depth [71, 4, 81, 30, 27, 86], dynamic channel width [32, 43, 29, 82] and dynamic routing [45] in convolutional neural networks. When it comes to the era of vision transformer, many works [77, 76, 69, 67, 48, 74, 57, 59, 58, 64] attempt to improve the inference efficiency by reducing the token redundancy. For instance, Liang *et al.* [48] identify and consolidate inattentive tokens into only one token in each transformer layer. Rao *et al.* [67] progressively discard less informative tokens between layers. Wang *et al.* [76] adjust the number of tokens to represent images. A more detailed literature review can be found in [28, 75]. Zhou *et al.* [96] propose assessing token significance with a ReLU function to effectively tackle point cloud analysis. Although these approaches have shown significant success in vision tasks, they require training a model from scratch or fine-tuning all parameters on the same dataset used in pre-training, making them unsuitable for efficient ViT adaptation.

In contrast to these methods, the proposed method can adapt pre-trained knowledge to diverse downstream datasets and tasks, while reducing the computational cost during inference by introducing negligible additional parameters.

## 3 ViTs Adaptation with Dynamic Tuning

In the following, we first introduce the vision transformer and the adapter architecture in Section 3.1. Subsequently, we present the proposed dynamic tuning (DyT) and explore its four variants in Section 3.2 and Section 3.3, respectively. Furthermore, we build an MoE-adapter, which effectively enhances the adaptation performance without introducing additional computational cost, as elaborated in Section 3.4. Finally, we introduce the loss functions of our method in Section 3.5.

### 3.1 Preliminary

Vision Transformer (ViT) consists of a stack of layers. The multi-head self-attention block denoted as $\mathrm{Attn}$, and multi-layer perceptron block, termed as $\mathrm{MLP}$ are the main components in each layer. The input of each layer can be denoted as $\mathbf{X} = [\mathbf{x}_{cls}, \mathbf{x}_1, \mathbf{x}_2, ...\mathbf{x}_N] \in \mathbb{R}^{(N+1) \times C}$, containing $N$ image tokens and a classification token $\mathbf{x}_{cls}$.

The adapter architectures have been widely explored in previous PEFT methods [12, 38, 33]. An adapter is generally designed as a bottleneck module, which consists of a project layer $\mathbf{W}_{down} \in \mathbb{R}^{C \times d}$ to squeeze the channel dimension and another project layer $\mathbf{W}_{up} \in \mathbb{R}^{d \times C}$ for dimension restoration. Here, $d$ denotes the bottleneck dimension and $d \ll C$ ensures the efficiency of the adapter. Given an input feature $\mathbf{X}$, the adapter operation is formulated as:

$$\mathbf{X}_{adapt} = \mathrm{Adapter}(\mathbf{X}) = \sigma(\mathbf{X}\mathbf{W}_{down})\mathbf{W}_{up}, \tag{1}$$

where $\sigma$ denotes a nonlinear function *e.g.* ReLU. In this study, we follow this general architecture to build the adapter within dynamic tuning. We place the adapter in parallel with an $\mathrm{Attn}$ block, $\mathrm{MLP}$, or an entire transformer layer, which will be presented in detail in Section 3.3.

### 3.2 Dynamic Tuning

In Vision Transformers (ViTs), the computational burden primarily resides in the transformer layers. Specifically, the operations in $\mathrm{Attn}$ and $\mathrm{MLP}$ account for approximately $35.8\%$ and $63.5\%$ of total FLOPs in ViT-B/16 [20]. Previous works [67, 48] have revealed that there exists a token-redundancy issue in ViTs and found that some tokens can be discarded without sacrificing the performance. Inspired by this, we propose an efficient ViT adaptation approach, named **Dynamic Tuning (DyT)**, which not only maintains parameter efficiency during fine-tuning but also reduces redundant computation during inference. The core idea is dynamically selecting tokens for processing within transformer blocks. Given the input tokens $\mathbf{X}$ of a block, DyT can be formulated as:

$$\mathbf{X}' = \mathrm{Block}(\mathrm{TD}(\mathbf{X})) + \mathrm{Adapter}(\mathbf{X}), \tag{2}$$

where $\mathrm{Block}$ can represent an multi-head self-attention block $\mathrm{Attn}$, a multi-layer perceptron $\mathrm{MLP}$, or an entire transformer layer. The proposed token dispatcher (TD) learns to selectively activate or deactivate tokens. Only the activated tokens are input into the $\mathrm{Block}$, while all tokens are processed by the $\mathrm{Adapter}$.

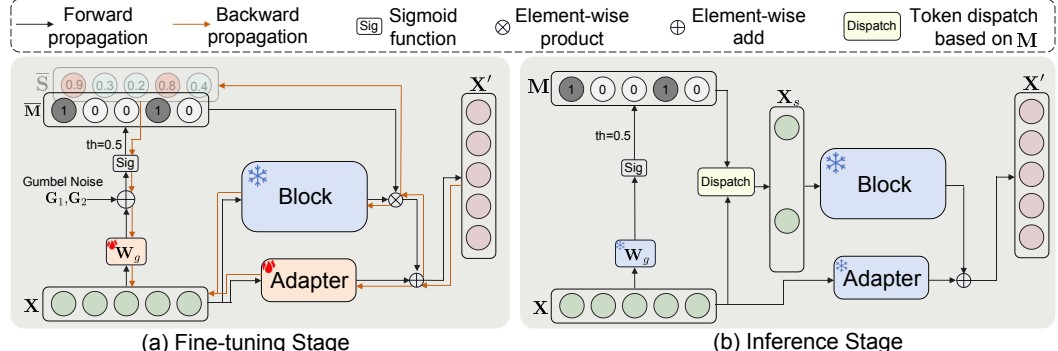

Figure 2: **Overview of Dynamic Tuning.** (a) In the fine-tuning stage, we adopt Gumbel Noise to enable end-to-end training. (b)In the inference stage, TD selects $K$ activated tokens $\mathbf{X}_s$ from $\mathbf{X}$ based on the mask $\mathbf{M}$, which saves the computations on those deactivated tokens in $\mathrm{Block}$. $\mathrm{Block}$ can represent a $\mathrm{Attn}$ block, a $\mathrm{MLP}$ block, or an entire transformer layer.

**Token dispatcher.** The key point in DyT is to select partial tokens which will be passed through $\mathrm{Attn}$ and/or $\mathrm{MLP}$. A straightforward approach is randomly selecting tokens with a predefined probability and adapting the model to conduct downstream tasks with these selected tokens. However, this simplistic strategy risks discarding informative tokens while retaining less important tokens, potentially hindering the adaptation performance. To tackle this problem, we propose a token dispatcher (TD), which learns to select tokens during the adaptation. Specifically, given the input tokens $\mathbf{X} \in \mathbb{R}^{(N+1) \times C}$, TD learns to selectively activate a sequential of tokens $\mathbf{X}_s \in \mathbb{R}^{K \times C}$, where $K$ represents the number of activated tokens. To achieve this, it should obtain a mask $\mathbf{M} \in \mathbb{R}^{N+1}$, which indicates whether a token should be activated or deactivated.

To obtain $\mathbf{M}$, we adopt a projection layer $\mathbf{W}_g \in \mathbb{R}^{C \times 1}$ followed by a sigmoid function to predict the activation probability $\mathbf{S} \in \mathbb{R}^{N+1}$. Then, we set 0.5 as the threshold to determine the activation of each token. This can be formulated as:

$$\mathbf{S} = \mathrm{Sigmoid}(\mathbf{X}\mathbf{W}_g), \ \mathbf{M}_n = \begin{cases} 1 & \text{if } \mathbf{S}_n \geq 0.5 \\ 0 & \text{if } \mathbf{S}_n < 0.5 \end{cases} \in \mathbf{M}. \tag{3}$$

$\mathbf{M}_n = 1$ denotes that the $n$-th token is activated and will subsequently undergo the process of $\mathrm{Block}$. Conversely, if $\mathbf{M}_n = 0$, the token will be deactivated and skip the $\mathrm{Block}$. In practice, we always set the mask value of the classification token $\mathbf{x}_{cls}$ to 1, allowing it to traverse the entire network. Notably, the number of additional parameters introduced in TD is negligible, with only $C$ parameters in the linear layer $\mathbf{W}_g$.

**Fine-tuning stage.** However, directly using threshold makes $\mathbf{M}$ a discrete decision, resulting in a non-differentiable problem during fine-tuning. To address this, we incorporate the Gumbel Noise [32] into sigmoid to replace the original sigmoid function *during fine-tuning*. It can be formulated as:

$$\overline{\mathbf{S}} = \mathrm{Gumbel\text{-}Sigmoid}(\mathbf{X}\mathbf{W}_g) = \mathrm{Sigmoid}(\frac{\mathbf{X}\mathbf{W}_g + \mathbf{G}_1 - \mathbf{G}_2}{\tau}), \tag{4}$$

where $\mathbf{G}_1, \mathbf{G}_2 \sim \mathrm{Gumbel}(0, 1)$. $\tau$ represents the temperature and is set to 5.0 by default. Further details of Gumbel-Sigmoid are provided in the Appendix A.7. Subsequently, we can obtain $\overline{\mathbf{M}}$ using the same operation in Equation.3. The Gumbel Noise makes the sampling of $\overline{\mathbf{M}}$ stochastic during training and we adopt $\overline{\mathbf{S}}$ as a differentiable approximation of $\overline{\mathbf{M}}$. Both of these help TD to be trained in an end-to-end manner. The calculation of forward and backward propagations during training can be formulated as:

$$\overline{\mathbf{X}}_s = \begin{cases} \mathrm{Block}(\mathbf{X}) \cdot \overline{\mathbf{M}} & \text{Forward Propagation} \\ \mathrm{Block}(\mathbf{X}) \cdot \overline{\mathbf{S}} & \text{Backward Propagation} \end{cases}, \tag{5}$$

The Equation 2 can be reformulated into:

$$\mathbf{X}' = \mathbf{X} + \overline{\mathbf{X}}_s + \mathrm{Adapter}(\mathbf{X}), \tag{6}$$

From Equation 5, only the block outputs, $\mathrm{Block}(\mathbf{X})$, of those activated tokens are retained, while others are masked out by $\overline{\mathbf{M}}$. As shown in Figure 2 (a), during the fine-tuning stage, all tokens within $\mathbf{X}$ still need to traverse the $\mathrm{Block}$.

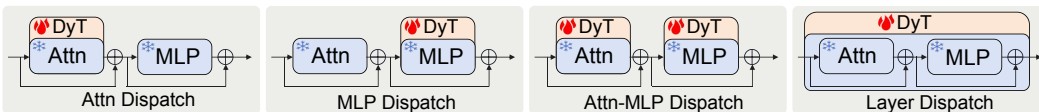

Figure 3: **Model variants.** For brevity, we omit the LayerNorm [1] in $\mathrm{Attn}$ and $\mathrm{MLP}$ blocks. "DyT" denotes the dynamic tuning presented in Figure 2.

**Inference stage.** During inference, we can directly adopt Equation 3 to generate the dispatch mask $\mathbf{M}$ and obtain activated tokens $\mathbf{X}_s \in \mathbb{R}^{K \times C}$ in $\mathrm{TD}$. Then, we can only feed them into $\mathrm{Block}$. Only processing $K$ tokens effectively reduces the computational cost because $K < N$. In practice, we add padding to the output from the $\mathrm{Block}$ to maintain tensor shape. This results in Equation 2. See Figure 2 (b) for a detailed illustration of the inference stage.

### 3.3 Model Variants

The $\mathrm{Block}$ in Equation 2 can be instantiated into any blocks in the original ViT, such as a multi-head self-attention block $\mathrm{Attn}$, a multilayer perceptron block $\mathrm{MLP}$, or even a complete transformer layer in ViT. Since the impact of skipping tokens in these blocks during the adaptation fine-tuning remains non-trivial to estimate and has not been explored in previous works, we propose four model variants and conduct experiments to determine the best practice.

- Attention Dispatch: Considering the quadratic computational complexity of the $\mathrm{Attn}$ block with respect to the token numbers, skipping tokens before applying $\mathrm{Attn}$ can significantly reduce computation. In this design, multi-head self-attention is exclusively performed between activated tokens, while other tokens are bypassed, which may hurt the interaction between tokens.

- MLP Dispatch: Based on the analysis in Section 3.2, we observe that $\mathrm{MLP}$ takes $\sim 63.5\%$ FLOPs in ViT-B/16 and propose to skip tokens only before $\mathrm{MLP}$. It enables that the interaction between tokens in $\mathrm{Attn}$ is not affected.

- Attention-MLP Dispatch: An alternative strategy is skipping tokens before both self-attention and MLP blocks. This design permits a higher activation rate in both $\mathrm{Attn}$ and $\mathrm{MLP}$ while maintaining similar computational efficiency comparable to "Attention Dispatch" and "MLP Dispatch". However, it costs double the additional parameters in adapters.

- Layer Dispatch: Inspired by "Attention-MLP Dispatch", we can dispatch tokens before a transformer layer. Specifically, tokens are identified by one $\mathrm{TD}$ to be activated or deactivated in the subsequent entire layer. With the same activation rate, its computation is similar to "Attention-MLP Dispatch" while requiring fewer parameters to build adapters.

We demonstrate the architecture variants in Figure 3. The experimental results and analyses of these variants are presented in Section 4.2.

### 3.4 MoE-Adapter

In DyT, the adapter is responsible for processing all tokens, requiring it to have enough capability, particularly when the downstream tasks *e.g.* semantic segmentation are challenging to adapt. To tackle this problem, we propose a MoE-adapter, inspired by mixture-of-experts [68, 80]. It effectively enhances the capability of the adapter with introducing negligible computation cost.

A MoE-adapter comprises a routing layer $\mathbf{W}_r \in \mathbb{R}^{C \times N}$ and $N$ adapter experts, denoted as $\{\mathbf{W}_{down}^i \in \mathbb{R}^{C \times d}, \mathbf{W}_{up}^i \in \mathbb{R}^{d \times C}\}_{i=1}^N$. The routing layer generates a series of scalar as the weights for the experts based on input features. The features from different images may yield distinct expert weights. Specifically, we first conduct average pooling across all tokens to generate a feature as the global representation of them. Subsequently, this representation is fed into the routing layer to generate weight scalars $\{\alpha^1, \alpha^2, ...\alpha^N\}$. Finally, tokens are processed by each expert independently and combined with the corresponding weight. We demonstrate this in Figure 4.

However, this increases the computational cost of the adapter in proportion to the number of experts $N$. To address this problem, we adopt its mathematical equivalence to process $\mathbf{X}$ in practice, which

can be formulated as:

$$\mathbf{X}_{adapt} = \sigma(\mathbf{X}\mathbf{W}_{down}^{moe})\mathbf{W}_{up}^{moe}, \tag{7}$$

where $\mathbf{W}_{down}^{moe} = \sum_{i=1}^{N} \alpha^i \mathbf{W}_{down}^i$ and $\mathbf{W}_{up}^{moe} = \sum_{i=1}^{N} \alpha^i \mathbf{W}_{up}^i$. The design endows the MoE-adapter with the same capacity as $N$ individual adapters, while maintaining computational efficiency equivalent to that of a single adapter (the computational cost in the routing layer is negligible).

### 3.5 Loss Functions

For an image $I$, we calculate the cross-entropy loss $\mathcal{L}_{cls} = -\mathbf{y}\log(\hat{\mathbf{y}})$ between the predicted probability $\hat{\mathbf{y}}$ and its corresponding one-hot label $\mathbf{y}$, to supervise the learning of classification. To control the average activation rate $r$ in the entire model, we add a loss term to constrain the number of activated tokens in the dynamic tuning, which can be formulated as: $\mathcal{L}_{rate} = (\frac{1}{L \times N} \sum_{l=1}^{L} \sum_{n=1}^{N} \mathbf{M}_n^l - r)^2$, where $L$ denotes the number of layers in ViT. $\mathbf{M}_n^l$ represents the mask generated in TD from the $l$-th layer. Additionally, we employ a loss $\mathcal{L}'_{cls} = -\mathbf{y}\log(\mathbf{y}')$ to supervise the adaptation of the complete model, where $\mathbf{y}'$ is the output probability without employing the token dis-patcher. Thus, this complete model can also act as a teacher to enhance the dynamic tuning during

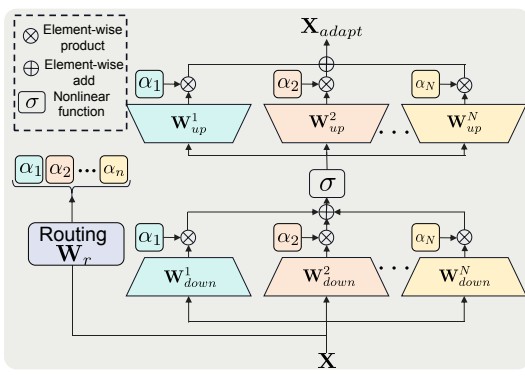

Figure 4: **The architecture of the MoE-adapter.** It is consist of $N$ adapter experts.

adaption by a distillation loss $\mathcal{L}_{distill} = \mathrm{KL}(y', y)$, where KL represents the Kullback-Leibler divergence loss. Therefore, the overall loss function is defined as $\mathcal{L} = \mathcal{L}_{cls} + \mathcal{L}'_{cls} + \mathcal{L}_{distill} + \alpha\mathcal{L}_{rate}$, where $\alpha$ serves as the weight of the activation rate loss and is set to 2.0 by default. Note that, DyT can also achieve competitive performance without the distillation loss (see Appendix A.6).

## 4 Experiments

### 4.1 Experiment Settings

**Datasets.** To evaluate the adaptation performance, we conduct experiments on VTAB-1K [89] benchmark. The training data in this benchmark is extremely limited, with only 1,000 training samples for each task. Different from existing PEFT works [12, 38, 39], which mainly focusing on VTAB-1K, we also conduct experiments on three image classification datasets with complete training sets, including CIFAR-100 [41], SVHN [24], Food-101 [6]. Additionally, we adopt two video datasets, Kinetic-400 (K400) [10] and Something-Something V2 (SSv2) [25], to evaluate the performance when the number of tokens scaled up. All images or frames are resize into 224 × 224. For the dense prediction task, we evaluate our method on two widely recognized semantic segmentation datasets, AED20K [93] and COCO-stuff [7]. The results of semantic segmentation are presented in the Appendix A.4. We run each task three times and report the averaged results. The error bars are small ($< 0.1$) and omitted for simplification.

**Implementation Details.** We conduct all experiments based on the ViT-B/16 [20], which is pre-trained on ImageNet21K [17] with full supervision, unless otherwise specified. Results based on ViT-L are presented in the Appendix A.10. The bottleneck dimension $d$ is set to 64 by default. We adopt the same training schedule as [12]. Detailed hyperparameters for each experiment can be found in the Appendix A.8. The default setting in experiments is marked in color.

### 4.2 Analysis

**Model variants.** In Table 1, we compare the performance of four model variants across both image and video datasets. We set the activation rate $r$ in "Attention Dispatch" and "MLP Dispatch" variants to 0.5, and to 0.7 for "Attention-MLP Dispatch" and "Layer Dispatch" variants, and train each respectively. This results in similar average FLOPs for four variants. We observe that the default

Table 1: **Comparison of model variants.** "Params. (M)" indicates the additional parameters in backbones. "FLOPs (G)" denotes the average FLOPs on CIFAR-100.

| Model Variant | Params.(M) ↓ | FLOPs (G) ↓ | Image Accuracy (%) ↑ | | | Video Accuracy (%) ↑ | |
|---|---|---|---|---|---|---|---|
| | | | CIFAR-100 | SVHN | Food-101 | K400 | SSv2 |
| Attention Dispatch | **1.19** | 14.77 | 84.58 | 96.55 | 86.67 | 69.67 | 41.56 |
| MLP Dispatch | **1.19** | **12.21** | **91.37** | **97.08** | **90.32** | **75.28** | **45.43** |
| Attention-MLP Dispatch | 2.38 | 12.93 | 90.03 | 96.66 | 88.61 | 74.62 | 41.83 |
| Layer Dispatch | **1.19** | 13.08 | 89.38 | 96.57 | 89.05 | 74.72 | 43.94 |

Table 2: **Effectiveness of MoE-Adapter.** DyT† denotes the DyT with MoE-adapters. Standard adapter is enough to handle image datasets while MoE-adapter is more suitable for challenging scenarios, such as video datasets. It theoretically does not increase extra computational cost, but the FLOPs slightly vary in different models since the learned token dispatch strategy in the TD is different. $N$ represents the number of experts.

| Model | Params. (M) ↓ | FLOPs (G) ↓ | Image Accuracy (%) ↑ | | | Video Accuracy (%) ↑ | |
|---|---|---|---|---|---|---|---|
| | | | CIFAR-100 | SVHN | Food-101 | K400 | SSv2 |
| DyT | 1.19 | 12.21 | **91.37** | **97.08** | **90.32** | 75.28 | 45.43 |
| DyT† $N = 2$ | 2.40 | 12.58 | 91.07 | 96.09 | 89.98 | 74.55 | 45.78 |
| DyT† $N = 4$ | 4.80 | 12.29 | 91.01 | 96.90 | 89.77 | **75.43** | **46.56** |
| DyT† $N = 8$ | 9.60 | 12.44 | 90.45 | 96.84 | 89.53 | 75.34 | 46.06 |
| DyT† $N = 12$ | 14.40 | 12.43 | 90.31 | 96.72 | 89.32 | 75.17 | 44.97 |

setting "MLP Dispatch" achieves superior performance across five datasets while maintaining the lowest computational cost. Although "Attention-MLP Dispatch" and "Layer Dispatch" also exhibit good performance on K400, the former incurs double additional parameters while the latter lacks generalization capability on other datasets. The comparison between "MLP Dispatch" and other variants proves that only skipping tokens in MLP blocks is a better design. More investigations on model variants can be found in our Appendix A.3.

**Effectiveness of MoE-adapter.** We conduct experiments to explore the effectiveness of the MoE-adapter and the results are illustrated in Table 2. The MoE-adapter design ensures that the FLOPs will theoretically remain the same as the ordinary adapter, with the computational cost from the routing function being negligible. However, in practical scenarios, the computational cost is also influenced by the learned token dispatch strategy within the Token Dispatcher (TD), leading to slightly varying FLOPs across different models in Table 2.

We observe that the performance on image datasets drops when we increase the expert number in MoE-adapter. This phenomenon can be attributed to the simplicity of image datasets and the model does not require too many parameters to adapt. In contrast, for video datasets, such as K400 and SSv2, the best accuracy is achieved 4 experts. The reason behind this is that the domain gap between the pre-training dataset and video datasets is large and the model needs sufficient adapter capacity to learn the adaptation. This proves that we can introduce the MoE-adapter when the target dataset is difficult to adapt.

**Visualization of token activation rate in each layer.** In Figure 5, we visualize the token activation rates across different layers in ViT-B/16 [20]. We observe that the model tends to activate more tokens in lower layers while deactivating tokens in higher layers. This phenomenon can be attributed to that the model tends to take more general knowledge from the lower layers of the pre-trained model and learn more task-specific knowledge in higher levels. Additionally, the activation results vary across different datasets. For instance, SSv2 demonstrates increased token activation rate in Layer 5 and Layer 6 compared to other datasets, whereas SVHN experiences substantial token deactivation in Layer 6, 7, and 8. This discrepancy arises from that the model needs different knowledge from the pre-trained weights to address dataset-specific challenges.

It is noteworthy that nearly all tokens are deactivated in the final layer across five datasets, especially CIFAR-100, SVHN, and K400, where the activation rates are exactly *0%*. This indicates that on these datasets, we can *directly drop* the original MLP block in Layer11 without hurting the performance, which further reduces about 4.7M [*], 5.4% of the ViT-B total parameters.

---

[*]There are two linear layers with weights of size $768 \times (4 \times 768)$ in a MLP block, results in $2 \times 768 \times (4 \times 768) \approx 4.7M$ parameters

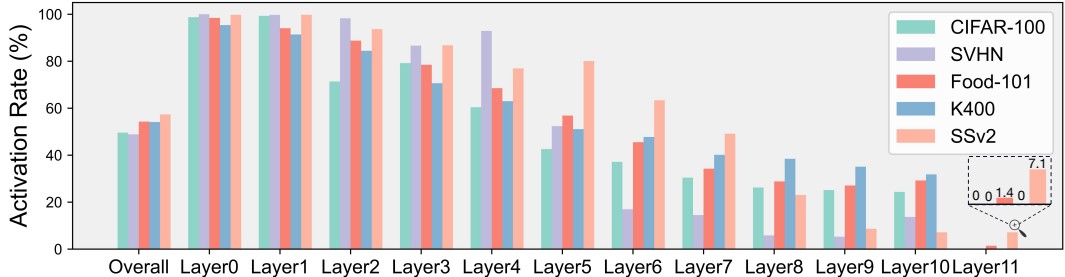

Figure 5: **Token activation rate in different layers.** We visualize the token activation rates in ViT-B/16. "Overall" denotes the mean activation rate in the whole model, which arrives at around 50% when $r$ is set to 0.5. "Layer0" and "Layer11" denote the lowest and highest level, respectively. Notably, the activation rate in the last layer is exactly *0%* on CIFAR-100, SVHN, and K400 datasets.

**Visualization of activated tokens.** In Figure 6, we visualize two representative samples from K400. We can observe that the model tends to deactivate those tokens that are less informative *e.g.* tokens of the sky in (a) and tokens of grass in (b). In higher layers, such as layer7 and layer10, only those tokens from the primary objects are activated. This further proves the the existence of token redundancy problems in ViT and provides validation for the rationality behind our approach. Additional visualizations are available in Appendix A.11.

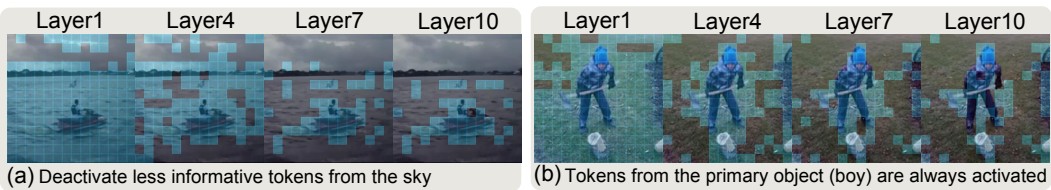

Figure 6: **Visualization of activated tokens.** We present two representative samples from the K400 dataset. **Blue patches** represent the tokens activated in token dispatcher (Detailed in Section 3.2). Results verify that the token dispatcher has learned to identify informative tokens during fine-tuning.

### 4.3 VTAB-1K Results

**Comparison with PEFT methods.** To evaluate the adaptation performance when the training data is limited, we adapt the VTAB-1K [89] benchmark, which a widely employed to evaluate the performance of adaptation tasks. Following exiting works, we reduce the bottleneck dimension $d$ to 8, resulting in only 0.16M extra parameters. We vary the activation rate $r$ in DyT in the range [0.5, 0.7, 0.9] and conduct fine-tuning, obtaining three models with different inference costs.

The results are presented in Table 3. Although previous methods, such as ConvPass [38] and Res-Tuning [37], achieve satisfactory performance, they do not improve the efficiency during inference and even introduce additional FLOPs compared with full fine-tuning. In contrast, with only 12.54 GFLOPs, about 71% of the cost in original ViT-B, our method has outperformed previous methods *e.g.* LoRA [34] and VPT [36], by a large margin. Remarkably, the DyT model with $r = 0.9$ does not surpass the performance of the DyT model with $r = 0.7$. This observation suggests the presence of computational redundancy in the transformer after adaptation, which further validates the significance of dynmaic tuning. These experimental results validate that DyT effectively improves parameter efficiency and inference efficiency while maintaining the adaptation performance.

**Dynamic tuning achieves actual acceleration.** As a hardware-agnostic metric, FLOPs is not the most suitable choice to evaluate the inference efficiency across diverse platforms. The real inference speed has usually been ignored in previous PEFT methods. Here, we adopt two GPUs (Tesla V100 and Tesla T4) and a CPU Xeon(R) Platinum 8163 to comprehensively evaluate the efficiency of our methods and three representative PEFT methods, including LoRA [34], AdaptFormer [12], and VPT [36]. The batch size during inference is set to 512 and 32 for GPUs and the CPU, respectively. The results, as summarized in Table 4, reveal that our method achieves better performance while effectively accelerating inference speed compared to existing PEFT methods on different platforms.

Table 3: **Performance and efficiency comparison on VTAB-1K**. "Group Mean" indicates the averaged accuracy of three groups. "Params. (M)" denotes the number of trainable parameters in **backbones**. "FLOPSs (G)" is the average FLOPs across all datasets. **Bold font** and underline denote the best and the second-best performance respectively.

| | ● Natural | | | | | | | ● Specialized | | | | ● Structured | | | | | | | | Group Mean | Params. (M) | FLOPs (G) |
|---|---|---|---|---|---|---|---|---|---|---|---|---|---|---|---|---|---|---|---|---|---|---|
| | CIFAR-100 | Caltech101 | DTD | Flowers102 | Pets | SVHN | Sun397 | Camelyon | EuroSAT | Resisc45 | Retinopathy | Clevr-Count | Clevr-Dist | DMLab | KITTI-Dist | dSpr-Loc | dSpr-Ori | sNORB-Azim | sNORB-Elev | | | |
| *Traditional methods* | | | | | | | | | | | | | | | | | | | | | | |
| Full tuning | 68.9 | 87.7 | 64.3 | 97.2 | 86.9 | 87.4 | 38.8 | 79.7 | 95.7 | 84.2 | 73.9 | 56.3 | 58.6 | 41.7 | 65.5 | 57.5 | 46.7 | 25.7 | 29.1 | 68.96 | 85.80 | 17.58 |
| Frozen | 63.4 | 85.0 | 63.2 | 97.0 | 86.3 | 36.6 | 51.0 | 78.5 | 87.5 | 68.6 | 74.0 | 34.3 | 30.6 | 33.2 | 55.4 | 12.5 | 20.0 | 9.6 | 19.2 | 57.64 | **0.00** | 17.58 |
| *Parameter-efficient tuning methods* | | | | | | | | | | | | | | | | | | | | | | |
| Adapter [33] | 69.2 | 90.1 | 68.0 | 98.8 | 89.9 | 82.8 | 54.3 | 84.0 | 94.9 | 81.9 | 75.5 | 80.9 | 65.3 | 48.6 | 78.3 | 74.8 | 48.5 | 29.9 | 41.6 | 73.85 | 0.16 | 17.61 |
| BitFit [87] | 72.8 | 87.0 | 59.2 | 97.5 | 85.3 | 59.9 | 51.4 | 78.7 | 91.6 | 72.9 | 69.8 | 61.5 | 55.6 | 32.4 | 55.9 | 66.6 | 40.0 | 15.7 | 25.1 | 65.21 | 0.10 | 17.58 |
| LoRA [34] | 67.1 | 91.4 | 69.4 | 98.8 | 90.4 | 85.3 | 54.0 | 84.9 | 95.3 | 84.4 | 73.6 | 82.9 | **69.2** | 49.8 | 78.5 | 75.7 | 47.1 | 31.0 | 44.0 | 74.60 | 0.29 | 17.58 |
| VPT [36] | **78.8** | 90.8 | 65.8 | 98.0 | 88.3 | 78.1 | 49.6 | 81.8 | 96.1 | 83.4 | 68.4 | 68.5 | 60.0 | 46.5 | 72.8 | 73.6 | 47.9 | 32.9 | 37.8 | 71.96 | 0.53 | 18.30 |
| SSF [38] | 69.0 | 92.6 | **75.1** | **99.4** | 91.8 | 90.2 | 52.9 | 87.4 | 95.9 | **87.4** | 75.5 | 75.9 | 62.3 | **53.3** | 80.6 | 77.3 | 54.9 | 29.5 | 37.9 | 75.69 | 0.20 | 17.58 |
| NOAH [91] | 69.6 | 92.7 | 70.2 | 99.1 | 90.4 | 86.1 | 53.7 | 84.4 | 95.4 | 83.9 | 75.8 | 82.8 | 68.9 | 49.9 | 81.7 | 81.8 | 48.3 | 32.8 | 44.2 | 75.48 | 0.36 | 17.58* |
| ConvPass [38] | 72.3 | 91.2 | 72.2 | 99.2 | 90.9 | **91.3** | 54.9 | 84.2 | **96.1** | 85.3 | 75.6 | 82.3 | 67.9 | 51.3 | 80.0 | **85.9** | 53.1 | **36.4** | 44.4 | 76.56 | 0.33 | 17.64 |
| AdaptFormer [12] | 70.8 | 91.2 | 70.5 | 99.1 | 90.9 | 86.6 | 54.8 | 83.0 | 95.8 | 84.4 | 76.3 | 81.9 | 64.3 | 49.3 | 80.3 | 76.3 | 45.7 | 31.7 | 41.1 | 74.75 | 0.16 | 17.61 |
| FacT-TT [39] | 71.3 | 89.6 | 70.7 | 98.9 | 91.0 | 87.8 | 54.6 | 85.2 | 95.5 | 83.4 | 75.7 | 82.0 | 69.0 | 49.8 | 80.0 | 79.2 | 48.4 | 34.2 | 41.4 | 75.30 | 0.04 | 17.58 |
| Res-Tuning [37] | 75.2 | 92.7 | 71.9 | 99.3 | **91.9** | 86.7 | **58.5** | 86.7 | 95.6 | 85.0 | 74.6 | 80.2 | 63.6 | 50.6 | 80.2 | 85.4 | 55.7 | 31.9 | 42.0 | 76.32 | 0.51 | 17.67 |
| *The proposed Dynamic Tuning* | | | | | | | | | | | | | | | | | | | | | | |
| DyT $r = 0.5$ | 73.6 | 94.8 | 73.0 | 99.1 | 91.4 | 87.0 | 56.4 | 87.3 | **96.1** | 85.6 | **76.7** | 82.8 | 63.8 | 52.7 | **83.7** | 83.6 | **57.3** | 34.6 | 44.3 | 77.14 | 0.16 | **12.54** |
| DyT $r = 0.7$ | 74.4 | **95.5** | 73.6 | 99.2 | 91.7 | 87.5 | 57.4 | **88.3** | 96.1 | 86.7 | **76.7** | **83.5** | 63.8 | 52.9 | 83.1 | **85.7** | 54.9 | 34.3 | 45.9 | **77.57** | 0.16 | 14.92 |
| DyT $r = 0.9$ | 74.3 | 94.9 | 73.1 | 99.2 | 91.4 | 87.8 | 57.1 | 87.9 | **96.1** | 85.9 | 76.0 | 83.3 | 64.8 | 51.5 | 83.4 | 84.0 | 54.8 | 35.1 | **46.4** | 77.30 | 0.16 | 17.07 |

\* NOAH cost larger than 17.58 FLOPS since it combines PEFT methods via neural architecture search.

Table 4: **Comparison of throughput.** "VTAB-1K Accuracy ↑" denotes the averaged accuracy of three dataset groups in VTAB-1K [89] benchmark.

| Method | VTAB-1K Accuracy ↑ | FLOPs (G) ↓ | V100 Throughput (img/s) ↑ | T4 Throughput (img/s)↑ | Xeon(R) 8163 Throughput (img/s) ↑ |
|---|---|---|---|---|---|
| Full tuning | 68.96 | 17.58 | 806.24 | 435.41 | 2.12 |
| LoRA [34] | 74.60 | 17.58 | 806.24 | 435.41 | 2.12 |
| AdaptFormer [12] | 74.75 | 17.61 | 767.30 | 400.42 | 1.97 |
| VPT [36] | 71.96 | 18.30 | 762.55 | 392.13 | 1.95 |
| DyT $r = 0.5$ | **77.14** | **12.54** | **912.30** | **524.93** | **3.89** |

**Comparison and compatibility with methods for efficient transformers.** We first investigate the domain adaptation performance of two representative methods DynamicViT [67] and EViT [48]. These methods are designed for efficient vision transformers. We adopt the optimal configurations outlined in their original papers and conduct experiments on VTAB-1K [89] benchmark. The results, as summarized in Table 5, reveal that both methods achieve high throughput *e.g.* > 1000 (img/s), while the performance is unsatisfying. Combining DynamicViT and EViT with AdaptFormer [12] results in performance improvements, validating the importance of exploring both parameter and inference efficiency for vision transformers. Despite these gains, DyT obviously surpasses them, highlighting the superiority of our approach.

Then, we explore the compatibility of our method with token pruning methods. Specifically, we combine DyT with ToMe [5], a training-free technique that progressively prunes tokens through token merging. From the results in Table 5, we find that ToMe can further enhance the throughput of DyT while maintaining accuracy. This proves the potential of our methods to be combined with existing token pruning methods *e.g.* [5, 11, 48]. Additionally, we apply ToMe to full tuning and AdaptFormer [12] in Table 3 and observe sub-optimal accuracy and throughput. These findings highlight that directly applying ToMe after fine-tuning or parameter-efficient fine-tuning is less effective compared to the proposed approach.

### 4.4 Further Exploration

**Effectiveness on image datasets with sufficient training data.** Although the results from VTAB-1K benchmark have proven the superiority of our approach, we extend our investigation to complete image datasets, to evaluate the adaptation performance with sufficient training data. We conduct experiments on 6 datasets including CIFAR-100 [41], SVHN [24], Food-101 [6], Air [56], Pet [61],

Table 5: **Comparison with efficient transformers.** The throughput is measured on a Tesla V100 GPU. "Params. (M) ↓ " denotes the number of trainable parameters in **backbones**.

| Method | VTAB-1K Accuracy ↑ | FLOPs (G) ↓ | Param. (M) ↓ | Throughput (img/s) ↑ |
|---|---|---|---|---|
| DynamicViT [67] | 60.10 | 14.05 | 88.70 | 1010.40 |
| DynamicViT+AdaptFormer[12] | 75.48 | 14.23 | 3.10 | 954.82 |
| EViT [48] | 67.42 | 11.62 | 85.80 | 1233.45 |
| EViT+AdaptFormer[12] | 74.63 | 11.77 | 0.16 | 1152.38 |
| Full tuning + ToMe [5] | 68.68 | 13.12 | 85.80 | 989.29 |
| AdaptFormer [12] + ToMe [5] | 74.30 | 13.29 | 0.16 | 941.70 |
| DyT $r = 0.5$ | 77.14 | 12.54 | 0.16 | 912.39 |
| DyT $r = 0.5$ + ToMe [5] | 76.60 | 9.85 | 0.16 | 1114.70 |

and Car [22]. The results are demonstrated in Table 6. We further explore a straightforward approach, "Dynamic-Full", which has a token dispatcher as the same in DyT and is fine-tuned with all parameters. We observe that its performance becomes unstable and drops significantly on some datasets *e.g.* CIFAR-100 and Food-101. This phenomenon may arise due to the potential adverse impact of dynamic dispatch on the pre-trained parameters during full-tuning adaptation, thereby validating the importance of DyT. In this data-sufficient scenario, although our method achieves performance slightly below that of full tuning and AdaptFormer, it brings a significant reduction in FLOPs.

**Scaling token counts from images to videos.** We conduct experiments on video datasets to show the performance when the number of tokens scaled up. The number of input frames is set to 8. For video tasks, similar to [85, 13], we employ a cross-attention layer and a query token to aggregate features from different frames. The video classification is conducted on the query token. Additional implementation details are provided in the Appendix A.8. We demonstrate the results in Table 6. Although DyT achieves slightly lower performance than AdaptFormer and LoRA, it costs obviously fewer FLOPs. DyT† containing four experts can achieve the best average accuracy with only cost 12.29 GFLOPs, which further verifies the superiority of our design.

Table 6: **Results on image and video tasks.** "Avg." denotes the average results from the corresponding task. The FLOPs are evaluated on CIFAR-100 and K400.

| Method | Params. ↓ (M) | FLOPs ↓ (G) | CIFAR-100 | SVHN | Food-101 | Air | Pet | Car | Avg. | FLOPs ↓ (G) | K400 | SSv2 | Avg. |
|---|---|---|---|---|---|---|---|---|---|---|---|---|---|
| | | | *Traditional methods* | | | | | | | | | | |
| Full tuning | 85.80 | 17.58 | 90.91 | 97.29 | 90.69 | **80.53** | 93.11 | **88.71** | **90.21** | 142.53 | 75.48 | 45.22 | 60.35 |
| Linear | **0** | 17.58 | 85.87 | 56.29 | 88.07 | 40.51 | 92.78 | 52.85 | 69.40 | 142.53 | 69.04 | 27.64 | 48.34 |
| Dynamic-Full | 85.80 | 12.24 | 83.43 | 96.90 | 84.46 | 62.50 | 75.16 | 70.48 | 78.82 | 107.67 | 74.63 | 44.94 | 59.79 |
| | | | *Parameter-efficient tuning methods* | | | | | | | | | | |
| AdaptFormer [12] | 1.19 | 17.81 | **92.03** | 97.23 | **90.84** | 78.23 | **94.46** | 87.33 | 90.02 | 144.39 | **75.53** | 45.36 | 60.45 |
| LoRA [34] | 1.19 | 17.58 | 91.42 | **97.36** | 90.48 | 76.32 | 93.62 | 87.00 | 89.36 | 142.53 | 75.48 | 45.62 | 60.55 |
| VPT [36] | 0.07 | 18.32 | 91.46 | 95.72 | 90.41 | 68.91 | 93.92 | 80.72 | 86.88 | 148.44 | 73.46 | 38.17 | 55.82 |
| | | | *The proposed Dynamic Tuning* | | | | | | | | | | |
| DyT | 1.19 | **12.21** | 91.37 | 97.08 | 90.32 | 78.92 | 94.45 | 87.80 | 89.99 | 108.31 | 75.28 | 45.43 | 60.36 |
| DyT† $N = 4$ | 4.80 | 12.29 | 91.01 | 96.90 | 89.77 | 78.27 | 93.85 | 87.61 | 89.56 | **105.45** | 75.43 | **46.56** | **60.98** |

## 5  Discussion and Conclusion

Previous methods for ViT adaptation primarily focus on improving the efficiency *during the adaptation*, thus aiming to reduce additional parameters. However, with the increasing size and computational cost of ViT models, the inference cost *after the adaptation* is becoming a heavy burden. In this paper, we unify both of these two problems into the efficiency problem for ViT adaptation and propose dynamic tuning (DyT) to tackle them simultaneously. We validate its performance and generalization across various tasks and datasets.

**Limitations and future works.** DyT is currently designed for vision tasks. We believe it would be interesting to combine vision backbones with large language models *e.g.* [72] to build efficient large multi-modal models *e.g.* [50, 97]. DyT could be further developed to be compatible with multi-modal models, which would be our future direction.

## Acknowledgements

This work was supported by Damo Academy through Damo Academy Research Intern Program. Yang You's research group is being sponsored by NUS startup grant (Presidential Young Professorship), Singapore MOE Tier-1 grant, ByteDance grant, ARCTIC grant, SMI grant (WBS number: A-8001104-00-00), Alibaba grant, and Google grant for TPU usage.

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

# A  Appendix

We organize our appendix as follows.

- In Section A.1, we present frequently asked questions along with their corresponding answers.
- In Section A.2, we detail the difference between our method and other previous works.
- In Section A.3, we present more analysis on the proposed method.
- In Section A.4, we report the results on semantic segmentation datasets.
- In Section A.5, we report the performance on object detection and instance segmentation.
- In Section A.6, we verify the effectiveness of complete model and distillation during adaption.
- In Section A.7, we provide the details and a formal proof related to the Gumbel-Sigmoid mechanism.
- In Section A.8, we presents the implementation details for each experiment.
- In Section A.9, we demonstrate the generalization capability of our method with Swin Transformer [53].
- In Section A.10, we investigate the impact of scaling up the model size to ViT-L [20].
- In Section A.11, we provide additional visualizations of activated tokens in our appraoch.

## A.1  Frequent Questions

**Why the proposed method outperforms traditional adapters?**    We list the explanations below:

- The dynamic architecture in DyT enhances generalization. It introduces a form of disturbance in the input data, akin to Dropout [70]. This mechanism is particularly crucial when training data is limited *e.g*. VTAB-1K.
- The distillation loss in DyT. We adopt the complete model as the teacher of the dynamic model, significantly enhancing performance. Such a self-distillation mechanism is only available in the dynamic architecture.
- Previous work [29] and DynamicViT also show dynamic architectures outperforming static models with fewer FLOPs.

**Why Table 6 shows that using MoE-adapter results in fewer FLOPs?**    It is possible that MoE-adapter results in slight fewer FLOPs. We list the explanations below:

- The FLOPs of DyT depend on learned token dispatcher during fine-tuning and may slightly fluctuate around the target FLOPs (controlled by $\mathcal{L}_{rate}$).
- The extra computational cost of the adapters and the MoE adapters is nearly equivalent.

Thus, a DyT model with the MoE-adapter may activate fewer tokens in the learned token dispatcher, resulting in slightly reduced FLOPs.

**Why the FLOPs of $N = 12$ are paradoxically lower than that of $N = 8$ in Table 2?**    Theoretically, the MoE-adapter with any number of experts should have similar FLOPs to the standard adapter. Meanwhile, the actual activation rate of DiT during inference depends on the learned token dispatcher after fine-tuning, resulting in slight fluctuations in FLOPs between different models. These explain why DyT $N = 12$ may have slight lower FLOPs than DyT $N = 8$.

## A.2  Difference with previous Works

We compare DyT with more previous works and demonstrate the differences between our approach and these methods.

**Difference with DynamicViT and EViT.** Both DynamicViT [67] and EViT [48] are token pruning methods, whereas DyT is a token skipping method. DynamicViT learns to retain $P\%$ tokens at certain layers *e.g.* 3th,6th 9th layers in ViT-B. EViT only keeps top-K attentive tokens and fuses the inattentive tokens at certain layers *e.g.* 4th,7th, and 10th layer in ViT-B. These methods primarily focuses on accelerating the model within the same dataset used for pre-training, whereas DyT aims to improve efficiency during cross-domain adaptation.

**Difference with DiffRate.** DiffRate [11] is a token compression method that performs token pruning and merging simultaneously.

- In the DiffRate [11], token pruning and merging is inherently data-independent. After training, a transformer layer prunes or merges the same number of tokens across all input data. In contrast, DyT is a data-dependent approach. The router in DyT learns to skip varying numbers of tokens before each MLP block based on the input data.

- The prune and merge operations in DiffRate do not preserve complete feature maps, presenting challenges for dense prediction tasks. Thus, DiffRate requires modifications to address these tasks. Conversely, with only performing token skipping, DyT maintains complete feature maps, allowing it to effectively handle dense prediction tasks without any modifications.

**Difference with ToMe.** ToMe [5] is a training-free technique that enhances inference efficiency by merging tokens based on similarity at each layer. DyT employs token skipping instead of merging and can be seamlessly integrated with ToMe to further improve efficiency.

**Difference with CoDA.** CoDA [42] is a PEFT method that can also improve the inference efficiency.

- The token selection strategy in the token dispatcher is different. CoDA selects top-K tokens in each layer to pass through while DyT adopt learnable dispatchers to select an appropriate number of tokens for each input.

- The target model is different. Although CoDA [42] also improve both the parameter and inference during adaptation, CoDA primarily focus on the language model *e.g.* T5 [66], while DyT is specifically designed for vision transformers.

- The block to conduct token skipping is different. In CoDA, tokens directly skip the whole layer. In DyT, we propose four model variants, explore their effectiveness, and find that skipping the MLP block is the most suitable for vision transformer.

**Difference with AdaMix.** AdaMix [73] also leverage a mixture of adapters, but it fuses all experts by weight averaging after training, resulting in an standard adapter akin to AdaptFormer [12]. In contrast, the proposed MoE-adapter employs a learning-based router to generate scalar weights for the experts based on input features, allowing features from different images to yield distinct expert weights.

## A.3 More Analysis

### A.3.1 Investigations of dispatch level and dispatch strategy

The proposed DyT performs dynamic dispatch at the token level using the token dispatcher (TD). In addition to token-level dispatch, we also investigate a sample-level dispatch, where all tokens within the selected samples are activated, while all tokens will be deactivated for other samples. To verify the importance of TD, we compare it against random dispatch, which randomly activates tokens or samples during both fine-tuning and inference. The experimental results are presented in Table 7.

Our observations reveal that token-level dispatch with TD consistently achieves superior performance across all datasets, except for a slight decrease compared to sample-level dispatch on the SVHN dataset. Notably, TD can achieve much better performance than the random dispatch strategy on both token-level and sample-level dispatch, particularly on video datasets K400 and SSv2, thereby validating the dispatch strategy learned in TD. Furthermore, the token-level dispatch surpasses the sample-level dispatch by a significant margin on most datasets, which demonstrates the importance of the finer-grained activation.

Table 7: **Investigations of dispatch level and dispatch strategy.** Combining the token-level dispatch and TD strategy results in the best performance. We consider two dispatch levels: "Token" (where dispatch is performed at the token-level) and "Sample" (where dispatch is conducted at the sample-level). "TD" and "Random" represent the learned dispatcher and random dispatch strategy, respectively.

| Dispatch Level | | Dispatch Strategy | | Image Accuracy (%) ↑ | | | Video Accuracy (%) ↑ | |
|---|---|---|---|---|---|---|---|---|
| Token | Sample | TD | Random | CIFAR-100 | SVHN | Food-101 | K400 | SSv2 |
| ✓ | | ✓ | | **91.37** | 97.08 | **90.32** | **75.28** | **45.43** |
| | ✓ | ✓ | | 90.70 | **97.29** | 89.8 | 71.14 | 44.09 |
| ✓ | | | ✓ | 89.38 | 96.79 | 86.39 | 68.27 | 40.13 |
| | ✓ | | ✓ | 87.15 | 97.07 | 85.26 | 68.18 | 39.79 |

### A.3.2 FLOPs-Accuracy curves of model variants.

In Figure 7, we further visualize the FLOPs-Accuracy curves of four model variants. We control the FLOPs by changing the activation rate $r$ for the fine-tuning stage. For "Attention Dispatch" and "MLP Dispatch", we explore the activation rate in the range [0.1, 0.3, 0.5, 0.7, 0.9]. To maintain similar FLOPs as "Attention Dispatch" and "MLP Dispatch", we adjust the activation rate for "Attention-MLP Dispatch" and "Layer Dispatch" within the range [0.5, 0.6, 0.7, 0.8, 0.9]. Across all datasets, "MLP Dispatch" consistently outperforms other variants. The performance of "Attn Dispatch" experiences a significant drop when the activation rate is lower than 0.9, which also indicates that skipping tokens for self-attention blocks in ViT is not a suitable way, due to the importance of the token mixing function of self-attention. Remarkably, "MLP Dispatch" can surpass full tuning with obviously fewer FLOPs on both CIFAR-100 and Food-101, further validating the effectiveness of our approach.

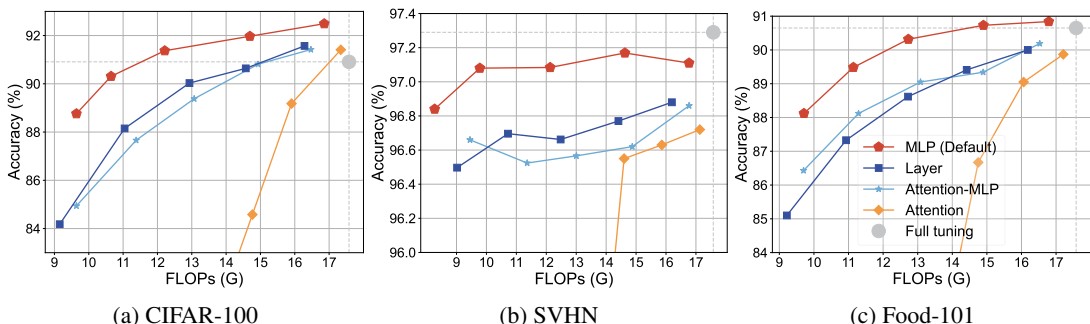

(a) CIFAR-100        (b) SVHN        (c) Food-101

Figure 7: **FLOPs-Accuracy curves of model variants on CIFAR100, SVHN, and Food-101 datasets.** "MLP Dispatch" achieve the best FLOPs-Accuracy trade-off. We explore the activation rate $r$ within [0.1, 0.3, 0.5, 0.7, 0.9] for "Attention Dispatch" and "MLP Dispatch". For "Attention-MLP Dispatch" and "Layer Dispatch" models, $r$ is adjusted within the range [0.5, 0.6, 0.7, 0.8, 0.9]. "Full tuning" denotes the traditional full fine-tuning approach. To conserve space, we use simplified names for the variants.

### A.3.3 Different bottleneck dimensions

We investigate the impact of the bottleneck dimensions of the adapter in DyT and the results are summarized in Table 8, revealing that different datasets prefer different configurations. When the dataset is easy to adapt, such as CIFAR-100, a bottleneck dimension of $d = 4$ is sufficient to achieve satisfactory performance. Conversely, video datasets require larger adapter dimensions *e.g.* $d = 256$ to attain better performance. We set the default bottleneck dimension to 64, which achieves a balance between performance and cost, such as the number of additional parameters and FLOPs. It is worth noting that the relationship between FLOPs and bottleneck dimension is not strictly monotonic due to the existence of token dispatcher. For instance, when $d = 16$, the FLOPs are lower than when $d = 1$, since the training may converge at activating fewer tokens as the number of adapter's parameters increases.

Table 8: **Different bottleneck dimensions.** Different datasets prefer different bottleneck dimensions. $d = 64$ strikes a balance between accuracy and cost, such as extra parameters and FLOPs. "FLOPs (G)" denotes the average FLOPs on CIFAR-100.

| Dimension | Params. (M) ↓ | FLOPs (G) ↓ | Image Accuracy (%) ↑ | | | Video Accuracy (%) ↑ | |
| --- | --- | --- | --- | --- | --- | --- | --- |
| | | | CIFAR-100 | SVHN | Food-101 | K400 | SSv2 |
| 1 | **0.03** | 12.11 | 91.46 | 95.5 | 89.65 | 71.78 | 36.22 |
| 4 | 0.08 | **11.89** | **91.57** | 96.61 | 89.99 | 73.16 | 39.24 |
| 16 | 0.30 | 12.06 | 91.46 | 96.98 | 90.24 | 73.65 | 42.49 |
| 64 | 1.19 | 12.21 | 91.37 | **97.08** | 90.32 | 75.28 | 45.43 |
| 256 | 4.73 | 13.32 | 91.18 | 97.03 | **90.42** | **75.33** | **46.73** |

### A.3.4 Investigation on the Temperature

We explore the temporal $\tau$ of the proposed dynamic tuning. The results are demonstrated in Table 9. When the temperature is smaller *e.g.* 0.1, the Gumbel Sigmoid tends to produce binary outputs close to 0 or 1. Conversely, larger temperatures lead to more uniform outputs, approaching 0.5. Results demonstrate that the performance is not too sensitive to the temperature and our model can achieve reasonable performance with all temperatures in the table. We also observe that the model with $\tau = 1$ achieves the best performance on CIFAR-100, SVHN, and SSv2, while decaying the temperature with a schedule leads to the best result on Food-101, which shows that adjusting the temperature can help the model to achieve better performance. Since identifying the optimal temperature is not the primary focus of this paper, we directly set the temperature to 5 by default.

Table 9: **Different temperature $\tau$ in dynamic tuning.** "Schedule" denotes that the temperature gradually decays from 5 to 0.1 during fine-tuning. The default setting is marked in color .

| Temperature | Image Accuracy (%) ↑ | | | Video Accuracy (%) ↑ | |
| --- | --- | --- | --- | --- | --- |
| | CIFAR-100 | SVHN | Food-101 | K400 | SSv2 |
| 0.1 | 90.91 | 96.24 | 89.72 | 73.16 | 44.84 |
| 1 | **91.61** | **97.20** | 90.08 | 74.38 | **45.69** |
| 5 | 91.37 | 97.08 | 90.32 | **75.28** | 45.43 |
| Schedule | 91.58 | 97.13 | **90.39** | 74.57 | 45.51 |

### A.4 Generalization in semantic segmentation

We also conduct experiments on two well-recognized semantic segmentation datasets ADE20K [93] and COCO-stuff [7] to demonstrate the ability of DyT on dense prediction tasks. Results are presented in Table 10. Following previous works [3, 19], we adopt the UperNet [78] as the segmentation head, and all images are resized into $512 \times 512$. We observe that the computational cost of semantic segmentation is much higher than the image and video discrimination tasks, primarily due to the high-resolution feature maps. Both DyT and DyT† still can reduce the computational cost obviously and achieve better performance than other PEFT methods, only slightly lower than full tuning on COCO-Stuff dataset.

### A.5 Generalization in Object Detection and Instance Segmentation

We conduct experiments on COCO [49] to explore the generalization of our method in object detection and instance segmentation. We adopt ViTDet [44] as the detector and fine-tune it for 12 epochs on the COCO dataset. The bottleneck dimension $d$ in adapters is set to 128. As shown in Table 11, DyT exhibits superior performance compared to AdapterFormer [12], with fewer FLOPs. Our MoE-adapter further enhances DyT without additional computational cost, validating the effectiveness of our design.

However, full tuning achieves the best performance, surpassing other methods significantly. This is likely due to the gap between bounding box regression and the pre-training of the vision transformer, necessitating more parameter updates in the backbone. This challenge motivates us to design more powerful PEFT methods and integrate them with DyT to reduce the performance gap with full tuning.

Table 10: **Results on semantic segmentation.** "Avg." denotes the average results from the corresponding two datasets. The FLOPs is measured ADE20K.

| Method | Params. ↓ | Semantic Segmentation Datasets | | | |
| | | FLOPs ↓ | mIOU | | |
| | (M) | (G) | ADE20K | COCO-stuff | Avg. |
|---|---|---|---|---|---|
| *Traditional methods* | | | | | |
| Full tuning | 85.80 | 605.36 | 47.66 | **45.89** | **46.78** |
| Linear | **0** | 605.36 | 43.86 | 44.46 | 44.16 |
| Dynamic-Full | 85.80 | 582.46 | 45.74 | 45.20 | 45.47 |
| *Parameter-efficient tuning methods* | | | | | |
| AdaptFormer [12] | 1.19 | 606.57 | 47.52 | 45.33 | 46.43 |
| LoRA [34] | 1.19 | 605.36 | 47.34 | 45.53 | 46.44 |
| VPT [36] | 0.07 | 606.36 | 46.38 | 44.87 | 45.63 |
| *The proposed Dynamic Tuning* | | | | | |
| DyT | 1.19 | 584.57 | 46.90 | 45.63 | 46.27 |
| DyT† $N = 4$ | 4.80 | **584.40** | **47.67** | 45.71 | 46.69 |

Table 11: **Results on object detection and instance segmentation.** We only measure the FLOPs in backbone and feature pyramid module.

| Method | Params. ↓ | COCO Datasets | | |
| | | FLOPs ↓ | BBox mAP | Seg mAP |
| | (M) | (G) | | |
|---|---|---|---|---|
| Full tuning | 85.80 | 554.86 | **44.67** | **39.78** |
| AdaptFormer [12] | 2.56 | 564.53 | 38.71 | 35.27 |
| DyT | 2.56 | 468.40 | 39.78 | 36.11 |
| DyT† $N = 4$ | 10.24 | 466.32 | 40.97 | 37.11 |

## A.6 Effectiveness of the Complete Model and Distillation

In the main paper, we adopt the complete model as our teacher, which does not employ a token dispatcher to skip tokens. Our proposed dynamic tuning allows the model to act as its own teacher during adaptation, a capability not achievable by other PEFT methods. We consider it to be a significant advantage of our approach. Joint training of the dynamic and complete models mitigates overfitting during adaptation, particularly with limited training data, such as VTAB-1K. Furthermore, the complete model, acting as a teacher, enhances the dynamic model's learning. These factors contribute to DyT's superior performance. Experimental results in Table 12 demonstrate that both complete model loss and distillation loss are useful for improving the performance of DyT. We also notice that introducing the complete model during training results $1.8 \times$ longer training time. Given that our primary contribution focuses on improving parameter and inference efficiency, the additional training time introduced by the complete model would be acceptable.

In Table 13, we further present the results across all datasets in detail. We can find that, DyT without $\mathcal{L}'_{cls} + \mathcal{L}_{distill}$ can still outperform most previous PEFT methods, further validating the superiority of our method.

## A.7 Details about Gumbel-Sigmoid

Gumbel-Softmax is proposed in [35] to conduct the differentiable sampling from a distribution. Given an unnormalized log probability † $\{\mathbf{E}_i\}_{i=1}^N$, the Gumbel-Softmax can be formulated as:

$$p_i = \frac{\exp\left(\left(\mathbf{E}_i + \mathbf{G}_i\right)/\tau\right)}{\sum_{n=1}^N \exp\left(\left(\mathbf{E}_n + \mathbf{G}_n\right)/\tau\right)}, \tag{8}$$

---

†The original definition in [35] assumes the input $\{\mathbf{E}_i\}_{i=1}^N$ to be *normalized log probability*, while practical implementations [65] demonstrate its effectiveness even with unnormalized inputs. We also follow [65] to formulate and implement the Gumbel-Softmax.

Table 12: **Effectiveness of loss functions** The default loss function in DyT is $\mathcal{L} = \mathcal{L}_{cls} + \mathcal{L}'_{cls} + \mathcal{L}_{distill} + \alpha\mathcal{L}_{rate}$. We gradually remove the complete model loss $\mathcal{L}'_{cls}$ and distillation loss $\mathcal{L}_{distill}$ from it, and find the performance drops.

| Method | VTAB-1K Accuracy ↑ | Training time |
|---|---|---|
| DyT | 77.14 | 1.8 × |
| DyT w/o $\mathcal{L}_{distill}$ | 76.70 | 1.8 × |
| DyT w/o $\mathcal{L}'_{cls} + \mathcal{L}_{distill}$ | 75.73 | 1.0 × |

Table 13: **Performance and efficiency comparison on VTAB-1K**. "Group Mean" indicates the averaged accuracy of three groups. "Params. (M)" denotes the number of trainable parameters in **backbones**. "FLOPSs (G)" is the average FLOPs across all datasets. **Bold font** and underline denote the best and the second-best performance respectively.

| | ● Natural | | | | | | | ● Specialized | | | | ● Structured | | | | | | | | Group Mean | Params. (M) |
|---|---|---|---|---|---|---|---|---|---|---|---|---|---|---|---|---|---|---|---|---|---|
| | CIFAR-100 | Caltech101 | DTD | Flowers102 | Pets | SVHN | Sun397 | Camelyon | EuroSAT | Resisc45 | Retinopathy | Clevr-Count | Clevr-Dist | DMLab | KITTI-Dist | dSpr-Loc | dSpr-Ori | sNORB-Azim | sNORB-Elev | | |
| *Traditional methods* | | | | | | | | | | | | | | | | | | | | | |
| Full tuning | 68.9 | 87.7 | 64.3 | 97.2 | 86.9 | 87.4 | 38.8 | 79.7 | 95.7 | 84.2 | 73.9 | 56.3 | 58.6 | 41.7 | 65.5 | 57.5 | 46.7 | 25.7 | 29.1 | 68.96 | 85.8 |
| Frozen | 63.4 | 85.0 | 63.2 | 97.0 | 86.3 | 36.6 | 51.0 | 78.5 | 87.5 | 68.6 | 74.0 | 34.3 | 30.6 | 33.2 | 55.4 | 12.5 | 20.0 | 9.6 | 19.2 | 57.64 | **0** |
| *Parameter-efficient tuning methods* | | | | | | | | | | | | | | | | | | | | | |
| Adapter [33] | 69.2 | 90.1 | 68.0 | 98.8 | 89.9 | 82.8 | 54.3 | 84.0 | 94.9 | 81.9 | 75.5 | 80.9 | 65.3 | 48.6 | 78.3 | 74.8 | 48.5 | 29.9 | 41.6 | 73.85 | 0.16 |
| BitFit [87] | 72.8 | 87.0 | 59.2 | 97.5 | 85.3 | 59.9 | 51.4 | 78.7 | 91.6 | 72.9 | 69.8 | 61.5 | 55.6 | 32.4 | 55.9 | 66.6 | 40.0 | 15.7 | 25.1 | 65.21 | 0.10 |
| LoRA [34] | 67.1 | 91.4 | 69.4 | 98.8 | 90.4 | 85.3 | 54.0 | 84.9 | 95.3 | 84.4 | 73.6 | 82.9 | **69.2** | 49.8 | 78.5 | 75.7 | 47.1 | 31.0 | 44.0 | 74.60 | 0.29 |
| VPT [36] | **78.8** | 90.8 | 65.8 | 98.0 | 88.3 | 78.1 | 49.6 | 81.8 | 96.1 | 83.4 | 68.4 | 68.5 | 60.0 | 46.5 | 72.8 | 73.6 | 47.9 | 32.9 | 37.8 | 71.96 | 0.53 |
| SSF [38] | 69.0 | 92.6 | **75.1** | **99.4** | 91.8 | 90.2 | 52.9 | 87.4 | 95.9 | **87.4** | 75.5 | 75.9 | 62.3 | **53.3** | 80.6 | 77.3 | 54.9 | 29.5 | 37.9 | 75.69 | 0.20 |
| NOAH [91] | 69.6 | 92.7 | 70.2 | 99.1 | 90.4 | 86.1 | 53.7 | 84.4 | 95.4 | 83.9 | 75.8 | 82.8 | 68.9 | 49.9 | 81.7 | 81.8 | 48.3 | 32.8 | 44.2 | 75.48 | 0.36 |
| ConvPass [38] | 72.3 | 91.2 | 72.2 | 99.2 | 90.9 | **91.3** | 54.9 | 84.2 | **96.1** | 85.3 | 75.6 | 82.3 | 67.9 | 51.3 | 80.0 | **85.9** | 53.1 | **36.4** | 44.4 | 76.56 | 0.33 |
| AdaptFormer [12] | 70.8 | 91.2 | 70.5 | 99.1 | 90.9 | 86.6 | 54.8 | 83.0 | 95.8 | 84.4 | 76.3 | 81.9 | 64.3 | 49.3 | 80.3 | 76.3 | 45.7 | 31.7 | 41.1 | 74.75 | 0.16 |
| FacT-TT [39] | 71.3 | 89.6 | 70.7 | 98.9 | 91.0 | 87.8 | 54.6 | 85.2 | 95.5 | 83.4 | 75.7 | 82.0 | 69.0 | 49.8 | 80.0 | 79.2 | 48.4 | 34.2 | 41.4 | 75.30 | 0.04 |
| Res-Tuning [37] | 75.2 | 92.7 | 71.9 | 99.3 | **91.9** | 86.7 | **58.5** | 86.7 | 95.6 | 85.0 | 74.6 | 80.2 | 63.6 | 50.6 | 80.2 | 85.4 | **55.7** | 31.9 | 42.0 | 76.32 | 0.51 |
| *The proposed Dynamic Tuning without $\mathcal{L}'_{cls} + \mathcal{L}_{distill}$* | | | | | | | | | | | | | | | | | | | | | |
| DyT $r = 0.5$ | 70.4 | 94.2 | 71.1 | 99.1 | 91.7 | 88.0 | 51.5 | 87.1 | 95.8 | 84.2 | 75.8 | 79.2 | 61.8 | 51.0 | 82.4 | 79.7 | 52.3 | 35.3 | 44.5 | 75.73 | 0.16 |
| DyT $r = 0.7$ | 73.9 | 94.9 | 72.1 | **99.4** | 91.8 | 88.4 | 55.5 | 87.2 | 95.6 | 86.2 | 75.9 | 80.3 | 61.8 | 51.7 | 83.1 | 81.6 | 53.7 | 35.3 | 45.2 | 76.69 | 0.16 |
| DyT $r = 0.9$ | 74.0 | 95.1 | 72.9 | 99.3 | 91.7 | 87.6 | 56.9 | 87.7 | 95.7 | 85.4 | 76.1 | 81.6 | 63.2 | 50.1 | 83.0 | 83.3 | 52.0 | 34.5 | 44.5 | 76.74 | 0.16 |
| *The proposed Dynamic Tuning with distillation* | | | | | | | | | | | | | | | | | | | | | |
| DyT $r = 0.5$ | 73.6 | 94.8 | 73.0 | 99.1 | 91.4 | 87.0 | 56.4 | 87.3 | **96.1** | 85.6 | **76.7** | 82.8 | 63.8 | 52.7 | **83.7** | 83.6 | **57.3** | 34.6 | 44.3 | 77.14 | 0.16 |
| DyT $r = 0.7$ | 74.4 | **95.5** | 73.6 | 99.2 | 91.7 | 87.5 | 57.4 | **88.3** | **96.1** | 86.7 | **76.7** | 83.5 | 63.8 | 52.9 | 83.1 | **85.7** | 54.9 | 34.3 | 45.9 | **77.57** | 0.16 |
| DyT $r = 0.9$ | 74.3 | 94.9 | 73.1 | 99.2 | 91.4 | 87.8 | 57.1 | 87.9 | **96.1** | 85.9 | 76.0 | 83.3 | 64.8 | 51.5 | 83.4 | 84.0 | 54.8 | 35.1 | **46.4** | 77.30 | 0.16 |

where $\mathbf{G}_i$ denotes the Gumbel Noise sampled from a Gumbel distribution ($\mathbf{G}_i \sim \text{Gumbel}(0, 1)$). We can consider the special case, where $N = 2$ and $E_2 = 0$, then $p_1$ can be defined as:

$$p_1 = \frac{\exp\left(\frac{\mathbf{E}_1 + \mathbf{G}_1}{\tau}\right)}{\exp\left(\frac{\mathbf{E}_1 + \mathbf{G}_1}{\tau}\right) + \exp\left(\frac{\mathbf{G}_2}{\tau}\right)} \qquad (9)$$

$$= \frac{1}{1 + \exp\left(-\frac{\mathbf{E}_1 + \mathbf{G}_1 - \mathbf{G}_2}{\tau}\right)} \qquad (10)$$

$$= \text{Sigmoid}\left(\frac{\mathbf{E}_1 + \mathbf{G}_1 - \mathbf{G}_2}{\tau}\right) \qquad (11)$$

$$= \text{Gumbel-Sigmoid}(\mathbf{E}_1), \qquad (12)$$

obtaining the formulation of Gumbel-Sigmoid. Researchers in previous works, such as [23, 51], have also leveraged the Gumbel-Sigmoid formulation to facilitate end-to-end training of neural networks.

## A.8 Implementation Details for each Task

**Experimental settings on VTAB-1K.** Following previous works [38, 39, 37], we fine-tune the model for 100 epochs on each dataset in VTAB-1K [89]. We *do not* use any data augmentation strategy in these experiments. We adopt the AdamW [55] optimizer. The learning rate is set to 1e-3 and gradually decays to 0 based on a cosine schedule [54].

**Experimental settings on complete image datasets.** We adopt the settings in Table 14 to fine-tune the ViT with the proposed dynamic tuning. Experiments on other parameter-efficiency methods such as AdaptFormer [12], LoRA [34], and VPT [36] also follow the settings in Table 14. When we train a model with full tuning, we adopt a 1/10 base learning rate to make the training stable, otherwise, the model can not achieve reasonable results.

Table 14: **Experimental settings for complete image datasets.** We present the hyperparameters in DyT. Following previous methods, we train the model with a 1/10 base learning rate in the full tuning setting. $lr = base\_lr \times batch\_size/256$

| | |
|---|---|
| Optimizer | AdamW [55] |
| Base learning rate | 1e-3 |
| Weight decay | 0.01 |
| Batch size | 1024 |
| Training crop size | 224 |
| Learning rate schedule | Cosine decay [54] |
| GPU numbers | 8 |
| Warmup epochs | 20 |
| Training epochs | 100 |
| Augmentation | RandomResizedCrop |

**Experimental settings on video datasets.** We adopt two video datasets, Kinetic-400 (K400) [10] and Something-Something V2 (SSv2) [25], to evaluate the performance as the token count scales up. The experimental settings are demonstrated in Table 15. Most of the settings are borrowed from [60]. The number of input frames is set to 8. We adopt multi-view during the test, which is a common practice in video action recognition. However, the original ViT lacks temporal modeling capabilities. To address this limitation, we draw inspiration from [85, 13]. By introducing a cross-attention layer after the ViT, along with a query token, we effectively aggregate temporal information across different frames. The final video action classification is performed based on this query token. Experiments on parameter-efficient fine-tuning methods also follow these settings. We adopt a 1/10 base learning rate for those experiments on full tuning.

Table 15: **Experimental settings for video datasets.** We follow most of settings in [60]. The number of input frames is set to 8 in all experiments. $lr = base\_lr \times batch\_size/256$

| Configuration | K400 [10] | SSV2 [25] |
|---|---|---|
| Optimizer | AdamW [55] | |
| Base learning rate | 1e-3 | |
| Weight decay | 0.01 | |
| Batch size | 128 | |
| Training Epochs | 12 | 50 |
| Training Resize | ShortSideJitter 224 - 256 | RandomResizedCrop |
| Training crop size | 224 | |
| Learning rate schedule | Cosine decay [54] | |
| Frame sampling rate | 16 | dynamic, evenly covering the whole video |
| Mirror | ✓ | ✗ |
| RandAugment [15] | ✗ | ✓ |
| Num. testing views | 1 spatial × 3 temporal | 3 spatial × 1 temporal |

**Experimental settings on semantic segmentation datasets.** We conduct experiments on ADE20K [93] and COCO-stuff [7] to demonstrate the ability of DyT on dense prediction tasks. ADE20K contains 20,210 images from 150 fine-trained semantic concepts. COCO-Stuff consists of about 164k images with 172 semantic classes. The experimental settings are demonstrated in Table 16. All

experiments for parameter-efficient fine-tuning methods also follow these settings. For the experiment on full tuning, we set adopt 1/10 learning rate for stable training and better performance.

Table 16: **Experimental settings for semantic segmentation datasets.** Following previous methods, we train the model with a 1/10 learning rate in the full tuning setting.

| Configuration | ADE20K [93] | COCO-stuff [7] |
|---|---|---|
| Optimizer | AdamW [55] | |
| Learning rate | 1e-3 | |
| Weight decay | 0.05 | |
| Batch size | 16 | |
| Training crop size | 512 | |
| Learning rate schedule | cosine decay [54] | |
| Training iterations | 160K | 80K |

## A.9 Generalization Capability of Transformer Architecture

To verify the generalization of the proposed dynamic tuning, we conduct experiments based on Swin-B [53]. The dynamic tuning can be directly applied to MLP blocks in the swin transformer without any modifications. The bottleneck dimension $d$ of the adapter in dynamic tuning also is set to 64. The results are demonstrated in Table 17. We can observe that the dynamic tuning can reduce both the tunable parameters and FLOPs while achieving comparable or even better performance across three datasets. This verifies the generalization capability of the dynamic tuning.

Table 17: **Generalization Capability on Swin Transformer [53]**. The experiments are conducted based on Swin-B. "Param. (M)" denotes the number of trainable parameters in **backbones**. "FLOPs (G)" denotes the average FLOPs on CIFAR-100. The bottleneck dimension $d$ is set to 64.

| Method | Params. (M) $\downarrow$ | FLOPs (G) $\downarrow$ | Image Accuracy (%) $\uparrow$ | | |
|---|---|---|---|---|---|
| | | | CIFAR-100 | SVHN | Food-101 |
| Full tuning | 86.74 | 15.40 | 91.82 | **97.66** | **93.05** |
| DyT $r = 0.1$ | 1.55 | 10.72 | 90.55 | 97.43 | 89.84 |
| DyT $r = 0.3$ | 1.55 | 12.07 | 91.26 | 97.38 | 90.66 |
| DyT $r = 0.5$ | 1.55 | 13.25 | 91.62 | 97.40 | 91.30 |
| DyT $r = 0.7$ | 1.55 | 14.05 | 92.14 | 97.21 | 91.96 |
| DyT $r = 0.9$ | 1.55 | 15.23 | **92.31** | 97.37 | 92.21 |

## A.10 Scaling-up Model Size

We verify the effectiveness of dynamic tuning when the model size is scaled up. We conduct experiments on ViT-L [20] and compare dynamic tuning with full tuning. The bottleneck dimension $d$ of the adapter in dynamic tuning is also set to 64. The results are demonstrated in Table 18. We can observe that with the activation rate set to 0.3, DyT has outperformed "full tuning" obviously on both CIFAR-100 and Food-101, while resulting in significantly lower computational cost.

We further compare the proposed dynamic tuning with full tuning on the VTAB-1K benchmark [89]. The results are demonstrated in Table 19. With only 0.44M tunable parameters and 43.52 GFLOPs, dynamic tuning surpasses full tuning across most datasets and achieves much better average performance.

Table 18: **Scale up the model size to ViT-L [20]**. "Param. (M)" denotes the number of trainable parameters in **backbones**. "FLOPs (G)" denotes the average FLOPs on CIFAR-100. The default setting is marked in color . The bottleneck dimension $d$ is set to 64.

| Method | Params. (M) ↓ | FLOPs (G) ↓ | CIFAR-100 | SVHN | Food-101 |
|---|---|---|---|---|---|
| Full tuning | 303.3 | 61.60 | 92.05 | **97.44** | 90.62 |
| DyT $r = 0.1$ | 3.17 | **32.56** | 91.26 | 97.04 | 89.98 |
| DyT $r = 0.3$ | 3.17 | 36.77 | 92.66 | 97.27 | 90.85 |
| DyT $r = 0.5$ | 3.17 | 43.79 | **93.49** | 97.38 | 91.49 |
| DyT $r = 0.7$ | 3.17 | 51.11 | 93.28 | 97.25 | **91.60** |
| DyT $r = 0.9$ | 3.17 | 60.05 | 93.44 | 97.23 | 91.59 |

Table 19: **Performance and efficiency comparison on VTAB-1K**. "Group Mean" indicates the averaged accuracy of three groups. "Full tuning" indicates fine-tuning all parameters. "Param. (M)" denotes the number of trainable parameters in **backbones**. "FLOPSs (G)" is the average FLOPs across all datasets. The bottleneck dimension is set to $d = 8$.

| | ● **Natural** | | | | | | | ● **Specialized** | | | | ● **Structured** | | | | | | | | ● **Structured** | | |
|---|---|---|---|---|---|---|---|---|---|---|---|---|---|---|---|---|---|---|---|---|---|---|
| | CIFAR-100 | Caltech101 | DTD | Flowers102 | Pets | SVHN | Sun397 | Camelyon | EuroSAT | Resisc45 | Retinopathy | Clevr-Count | Clevr-Dist | DMLab | KITTI-Dist | dSpr-Loc | dSpr-Ori | sNORB-Azim | sNORB-Elev | Group Mean | Param. (M) | FLOPs (G) |
| Full tuning | 69.5 | 96.2 | 73.8 | 98.8 | 90.7 | 91.6 | 44.8 | 85.8 | 96.2 | 87.8 | 75.3 | 83.0 | 62.0 | 50.8 | 80.0 | 85.8 | 54.6 | 29.7 | 35.4 | 75.7 | 303.3 | 61.60 |
| DyT $r = 0.5$ | 79.1 | 95.6 | 74.5 | 99.5 | 92.6 | 90.8 | 59.3 | 86.9 | 96.6 | 87.2 | 76.5 | 84.5 | 62.9 | 53.3 | 83.5 | 88.4 | 57.3 | 38.7 | 44.6 | 78.5 | 0.44 | 43.52 |

## A.11 Additional Visualizations of Activated Tokens

We provide more visualizations of activated tokens from samples in K400 [10] and SSv2 [25] in Figure 8 and Figure 9, respectively. Results demonstrate that most activated tokens in higher layers *e.g.* Layer10 come from the primary objects. This proves that the proposed token dispatcher learns to activate informative tokens.

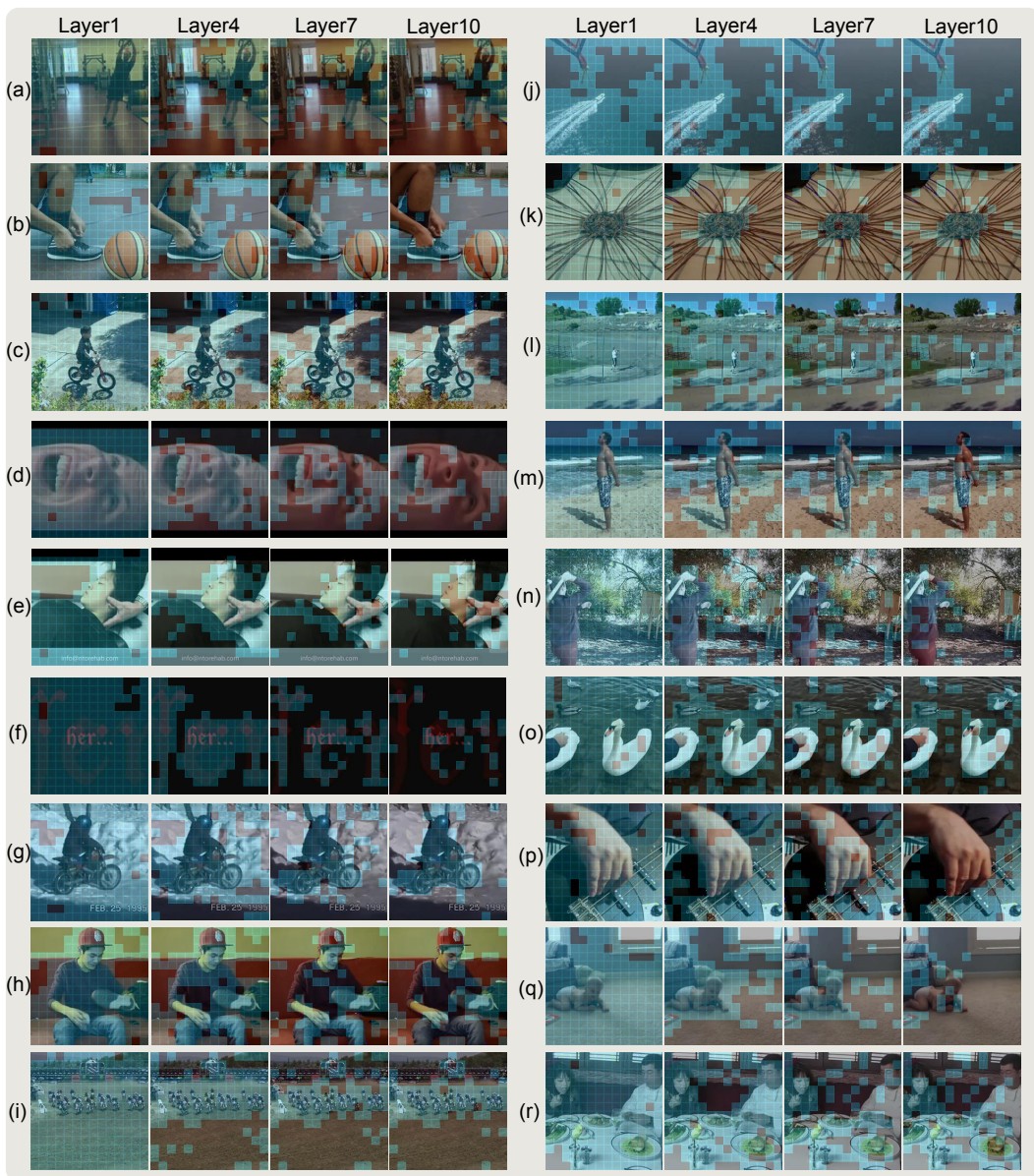

Figure 8: **Visualization of activated tokens.** We present representative samples from the K400 [10] dataset. **Blue patches** represent the tokens activated in token dispatcher. Results verify that the token dispatcher has learned to identify informative tokens during fine-tuning. Zoom in for better view.

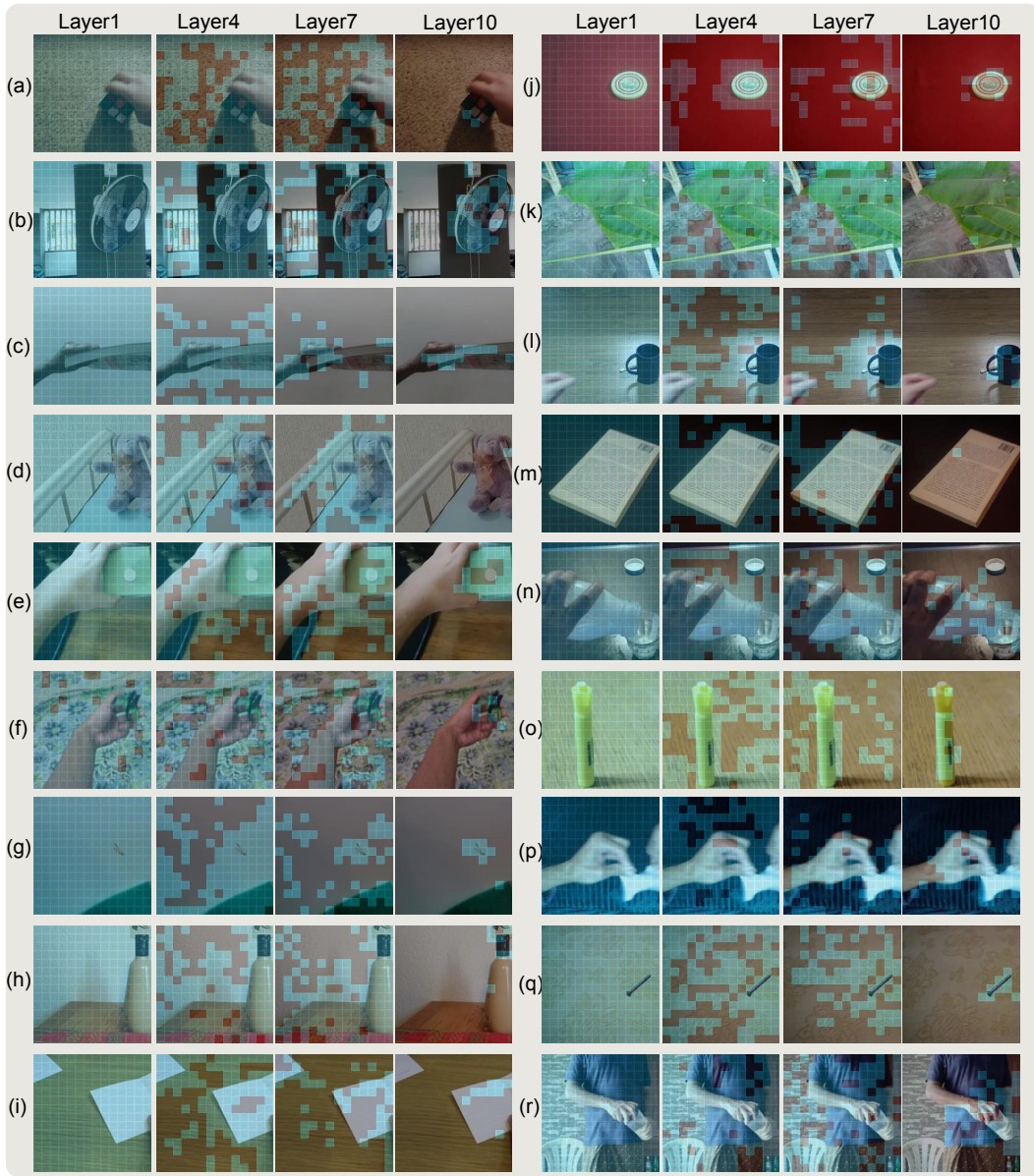

Figure 9: **Visualization of activated tokens.** We present representative samples from the SSv2 [25] dataset. **Blue patches** represent the tokens activated in token dispatcher. Results verify that the token dispatcher has learned to identify informative tokens during fine-tuning. Zoom in for better view.

