# OpenReview forum: "Dynamic Tuning Towards Parameter and Inference Efficiency for ViT Adaptation"
_NeurIPS.cc/2024/Conference — NeurIPS 2024 poster_

### Official Review · Reviewer_gv1w · 2024-07-08

**Soundness:** 3
**Presentation:** 3
**Contribution:** 2
**Rating:** 5
**Confidence:** 5

**Summary:**

This paper proposes DyT, which is a parameter-efficient fine-tuning method that can also achieve inference efficiency. DyT contains token dispatchers, which let tokens dynamically skip the original block, and MoE-adapter which uses multiple adapters to compose a mixture-of-experts. The paper evaluates the proposed method on various vision benchmarks to demonstrate its effectiveness and efficiency.

**Strengths:**

- This paper provides sufficient experimental results on image and video tasks.
- The problem is very meaningful. The mainstream PEFT methods achieve efficiency in training and storage, but cannot achieve efficiency in inference.

**Weaknesses:**

- My main concern is the novelty of the proposed method. DyT mainly includes two innovations: the token dispatcher and the MoE-Adapter. However, to my knowledge, both of these have been explored in earlier works.
    - First, token dispatcher is very similar to [1], both of them use a linear router to determine which tokens are input into the transformer's block and use parallel adapters to process all tokens. The difference lies in that [1] uses a soft top-k function to generate the mask, whereas this paper employs the sigmoid function.
    - Second, [2] also used MoE as the adapters. The difference is that [2] performs routing separately for the down projection and up projection, whereas in this paper, the two are bound together.

- The line spacing is too narrow.


[1] Conditional Adapters: Parameter-efficient Transfer Learning with Fast Inference. In NeurIPS 2023.

[2] AdaMix: Mixture-of-Adaptations for Parameter-efficient Model Tuning. In EMNLP 2022.

**Questions:**

- Please clarify the novelty of the proposed methods over Conditional Adapters and AdaMix.
- DyT employs a more complex loss function and distillation. Does this increase the training time?

**Limitations:**

The paper has discussed the limitations of DyT.

---

> ### Author Rebuttal · Authors · 2024-08-06
>
> > #### W1 & Q1: Clarify the novelty of the proposed methods over Conditional Adapters and AdaMix
>
> Thanks. We would like to clarify the novelty of our method over Conditional Adapters and AdaMix as follows:
>
> **Conditional Adapters[1]:**
> - The token selection strategy in the token dispatcher is novel. The token dispather in conditional adapter[1] selects top-K tokens in each layer to pass through. However, the token dispatcher in DyT **learn to select** an approprate number of tokens in each layer based on inputs.
> - The target model is different. The conditional adapter[1] is primarily designed for the encoder-decoder langauge model T5 [3]. Our experiments below demonstrate that applying this approach to vision transformers is suboptimal.  DyT’s novelty lies in proposing a token dispatcher specifically beneficial for vision transformer.
> - The block to conduct token skipping is novel. In conditional adapter [1], tokens directly skip the whole layer. In DyT, we propose four model variants, explore their effectiveness, and find that skipping the MLP block is suitable for vision transformer.
>
> To verify that the superiority of our token dispatcher in vision transformer, we conduct experiments by replacing the proposed token dispatcher with the soft top-K method from [1].
>
> **Setting:** DyT-top-K denotes using the soft top-K in [1] as the token dispatcher. We report the average FLOPs on CIFAR-100.
>
> | Method     | FLOPS | CIFAR-100     |   SVHN | Food-101
> | -------- | -------- | -------- | -------- | --------
> | DyT      | 12.21 |  91.37 | 97.08  | 90.32
> | DyT-top-K | 12.26 | 81.4   | 93.57  | 79.47
>
> **Analysis:** DyT achieves obviously better performance than "DyT-top-K". There are two reasons:
>
> - The soft top-K operation in [1] always skips K tokens per block. In contrast, DyT’s **learned token dispatcher** skips **fewer tokens in lower layers and more in higher layers** (see Figure 5). This indicates that the number of skipped tokens should vary by layer, and skipping a fixed number of tokens, as in [1], is suboptimal.
> - The computational allocation for each test sample is constant in [1], whereas DyT **allocates different FLOPs for different samples**, allowing the model to handle hard and easy samples differently.
>
> **Conclusion:**
> The token dispatcher in [1] is not suitable to vision transformer, verifing the importance of DyT. We will add this experiment into the Appendix in the revision.
>
> **AdaMix [2]:**
> - Our routing mechanism is novel. During training, AdaMix [2] lacks a **learning-based router**, instead randomly selecting a down projection layer and an up projection layer for each batch of data.
> Consequently, all images in a batch are processed by the same experts. In contrast, the MoE-adapter in DyT employs a learning-based router to generate scalar weights for the experts based on input features, allowing features from different images to yield distinct expert weights.
> - The adapter architecture is novel. During inference, AdaMix fuses all experts by weight averaging, resulting in an ordinary adapter like [4], whereas we retain all experts. Thus, the MoE-adapter in DyT has more capacity to tackle challenging tasks. Experiments below verify this.
>
> Below, we conduct experiments by replacing our MoE-adapter with AdaMix.
>
> **Setting:**
> "DyT-AdaMix" denotes using the AdaMix [1] as the adapter. "DyT $N=4$" denotes the DyT with our MoE-Adapter.
> We report the accuracy on three image datasets and two video datasets. The average FLOPs on CIFAR-100 and K400. The bottleneck dimension in the adapter is set to 64 for all models.
> | Method    | Params. (M) | Image FLOPs | CIFAR-100     |   SVHN | Food-101 | Video FLops|K400 | SSv2
> | -------- | -------- | -------- | -------- | -------- | -------- | -------- | -------- | --------
> | DyT      | 1.19 | 12.21  | 91.37 |  97.08 | 90.32 | 108.31  |75.28 | 45.43
> | DyT $N=4$ | 4.80 |12.29 |  91.01 | 96.90 | 89.77 | 105.45 | 75.43 | 46.56
> | DyT-AdaMix  | 1.19 |12.24 |  91.57 | 97.26 | 89.99 | 108.45 | 72.70 | 41.41
>
> **Analysis:**
> - "DyT-AdaMix" exhibits slightly better performance on image datasets due to its training strategy.
> - “DyT-AdaMix” does not enhance DyT’s performance on challenging tasks, such as video classification, and even significantly reduces performance.  This may be attributed to the random selection mechanism, which further diffuses learning on challenging tasks.
> - As an method to improve performance in challenging tasks without introducing extra FLOPs, our MoE-adapter achieves the best performance on K400 and SSv2.
>
> **Conclusion:**
>
> The adapter in AdaMix [2] is **a training strategy** rather than an effective adapter architecture. It does not assist DyT in tackling challenging tasks. We will add this experiment to the Appendix in the revision.
> > #### W2: The line spacing is too narrow.
>
> We sincerely appreciate this valuable suggestion. We find that author are permitted to add one page to the main paper in the final version. We will increase the line spacing as recommended.
>
> > #### Q2: DyT employs a more complex loss function and distillation. Does this increase the training time
>
> Thanks. We employ the complete model loss $L^\prime_{cls}$ and distillation loss $L_{distill}$ to improve the performance.  Although their use increases the duration of a training iteration by 1.8 times compared to not using them, this approach brings significant performance improvements (See Table 11). Given that our primary objective is to improve inference efficiency, this additional training cost is acceptable. Therefore, we believe it is valuable to include these losses during fine-tuning.
> > #### Reference
>
> [1] Conditional Adapters: Parameter-efficient Transfer Learning with Fast Inference. 2023.
>
> [2] AdaMix: Mixture-of-Adaptations for Parameter-efficient Model Tuning, 2022.
>
> [3] Exploring the Limits of Transfer Learning with a Unified Text-to-Text Transformer. 2020.
>
> [4] Adaptformer: Adapting vision transformers for scalable visual recognition, 2022.

---

> > ### Comment · Reviewer_gv1w · 2024-08-11
> >
> > Thank you for your response.
> >
> > My initial concerns were primarily about the novelty, as the designs of the token dispatcher and MoE adapter have overlap with existing work. After reading others' opinions, I realized that the mechanism of token dispatcher also bears a resemblance to  Dynamic ViT and token pruning methods. The authors’ response clarified the differences between these works, specifically:
> >
> > Compared with conditional adapter
> > + "DyT selects an appropriate number of tokens in each layer, while the conditional adapter selects a fixed number."
> >   + It is indeed different in this regard.
> > + "Conditional adapter is primarily designed for encoder-decoder models, while DyT is for ViT."
> >   + Although experiments were conducted only on the encoder-decoder model, the conditional adapter is not limited to a specific model architecture, just like LoRA.
> > + "Conditional adapter skips the whole layer, while DyT skips the MLP blocks."
> >   + Similarly, although the placement is different, this alone does not provide sufficient novelty unless a theoretical or intuitive motivation for doing so can be provided.
> >
> > Compared with AdaMix
> > + "AdaMix lacks a learning-based router and fuses experts"
> >   + There is still a lot of work being done on MoE adapters. For example, MoLoRA [1] uses a learnable router as an adapter and does not fuse experts.
> >
> > Compared with DynamicViT and token compression
> > + "DynamicViT and token compression methods prune a certain number or percentage of tokens in fixed layers, and cannot complete feature maps"
> >   + This issue existed in early token compression work, but later efforts, such as DiffRate, have addressed it. Combining token compression methods with existing PEFT approaches, such as AdaptFormer and LoRA, could serve as a strong baseline.
> >
> > In summary, I appreciate the authors’ response and acknowledge the additional experiments that demonstrate the effectiveness of DyT. However, the issues mentioned above still lead me to believe that DyT lacks sufficient innovation. Therefore, I will keep my score unchanged.
> >
> >
> > [1] Pushing Mixture of Experts to the Limit: Extremely Parameter Efficient MoE for Instruction Tuning. ICLR 2024
> >
> > [2] DiffRate : Differentiable Compression Rate for Efficient Vision Transformers. ICCV 2023

---

> > > ### Author Response · Authors · 2024-08-11
> > >
> > > Dear Reviewer gv1w:
> > >
> > > We appreciate the reviewer’s feedback and would like to address the points raised below:
> > >
> > > ## Part1/2
> > >
> > > **Compared with conditional adapter**
> > > |Question|Answer
> > > |-|-
> > > |"DyT selects an appropriate number of tokens in each layer, while the conditional adapter selects a fixed number."It is indeed different in this regard.| We appreciate that the reviewer acknowledges this contribution. The soft-topK token dispatcher serves as a core component in 'conditional adapter', which is indeed different from our approach. This verifies the distinction between our approach and 'conditonal adapter'.
> > > |"Conditional adapter is primarily designed for encoder-decoder models, while DyT is for ViT." Although experiments were conducted only on the encoder-decoder model, the conditional adapter is not limited to a specific model architecture, just like LoRA.| The 'conditional adapter' can be applied into ViT but does not work well. In our response, we conduct experiments and apply the 'conditional adapter' to ViT by replacing our token dispatcher with the soft-topK operation. However, DyT outperforms the 'conditional adapter' significantly, emphasizing the significance of our approach in ViT.
> > > |"Conditional adapter skips the whole layer, while DyT skips the MLP blocks." Similarly, although the placement is different, this alone does not provide sufficient novelty unless a theoretical or intuitive motivation for doing so can be provided.|In Vision Transformers (ViT), the self-attention and MLP blocks serve distinct roles in token mixing and feature projection, respectively.However, the impact of token skipping at different blocks on adaptation performance remains unexplored in previous works. Its potential impacts encourage us to explore. In our main paper (Lines 155-178), we analyze four variants, carefully considering their advantages and disadvantages. Additionally, in Figure 6 (Appendix), we conduct experiments to explore this and demonstrate the superiority of the “MLP dispatch” method. These analyses and experiments underscore **the non-trivial nature of the placement of token skipping** and further validate the significance of the exploration in DyT.
> > >
> > > **Compared with AdaMix**
> > > |Question|Answer
> > > |-|-
> > > |"AdaMix lacks a learning-based router and fuses experts" There is still a lot of work being done on MoE adapters. For example, MoLoRA [1] uses a learnable router as an adapter and does not fuse experts.| The MoLoRA [1] is adopted to reduce the additional parameters when converting a dense model into a Mixture of Experts (MoE) model. Our MoE-adapter aims to improve the performance of DyT on challenging tasks, while avoiding introducing additional computational cost. Notably, our MoE-adapter adopts weighted summation at both down-projection $W_{down}$ and up-projection $W_{up}$ in adapters, in contrast to MoLoRA [1], which aggregates all LoRA blocks. Hence, our architecture is also different from MoLoRA [1]. While MoE is a widely used and effective technique in previous works, our MoE-adapter offers a unique design and is **tailored to improve the performance of DyT**. Finally, we would like to clarify that we pay more attention to achieving both parameter and inference efficiency for ViT adaptation and the MoE-adapter is **not the main contribution** in our work.
> > >
> > > **Compared with DynamicViT and token compression**
> > > |Question|Answer|
> > > |-|-
> > > |"DynamicViT and token compression methods prune a certain number or percentage of tokens in fixed layers, and cannot complete feature maps" This issue existed in early token compression work, but later efforts, such as DiffRate, have addressed it. Combining token compression methods with existing PEFT approaches, such as AdaptFormer and LoRA, could serve as a strong baseline.| In the Diffrate model, token pruning is **inherently data-independent**. After training, a transformer layer **prunes or merges the same number of tokens across all input data**. Additionally, the prune/merge operation **can not preserve complete feature maps**, which poses challenges for dense prediction tasks. In comparison, **DyT is a data-dependent approach**. The router in DyT learns to skip varying numbers of tokens before each MLP block based on the input data. The visualization in Figure 8 and Figure 9 (Appendix) demonstrate that the learned token skipping strategy in DyT **selects different numbers of tokens to skip based on different inputs**, which is more flexible and reasonable. By avoiding prune/merge operations and only conducting token skipping, DyT maintains complete feature maps, enabling it to **effectively handle dense prediction tasks**. Furthermore, in our response to Reviewer fmWb, we also explore combining token compression methods with AdaptFormer together, and find that our method can surpass them obviously, verifying the superiority of designs in DyT.

---

> > > ### Author Response · Authors · 2024-08-11
> > >
> > > ## Part2/2
> > >
> > > In addition to distinguishing DyT from existing works, we summarize the contribution of our work below:
> > > - **Verify the feasibility:** The proposed method DyT proposes to learn a data-dependent architecture in a parameter-efficient manner during the ViT adaption. It verifies the feasibility of achieving both parameter and inference in ViT adaptation, which may direct an avenue for efficient fine-tuning and adaptation.
> > > - **Elaborate design:** To achieve better efficiency and performance, DyT explores the optimal placement of token skipping (Section 3.3 Model variants), through extensive analysis and experiments. We also design a MoE-adapter to improve the performane of DyT in challenging tasks.
> > > - **DyT can handle various adaptation tasks:**. DyT achieves both parameter and inference efficiency in various tasks, including image classification (Table3 and Table6), video classification (Table 6), semantic segmentation (Table 10), and object detection & instance segmentation (Our response to Reviewer Reviewer uJS2).

---

> ### Comment · Reviewer_gv1w · 2024-08-11
>
> Thank you for your response. I will carefully consider your clarifications, and if other reviewers also find the innovation of the paper to be sufficient during discussion, I will raise my score.

---

> > ### Author Response · Authors · 2024-08-11
> >
> > Dear Reviewer gv1w,
> >
> > Thanks for the reply and for considering raising the score for our work. We will discuss more with other reviewers in the next few days.  We will also sync the results with you. Thanks again for your time and effort in our work.
> >
> >
> > Best wishes,
> >
> > Authors

---

> > > ### Comment · Reviewer_gv1w · 2024-08-14
> > >
> > > Taking into account the discussions between the author and the other reviewers, I have raised the score. However, I would like the author to include a discussion of the aforementioned work in the final version.

---

> > > > ### Author Response · Authors · 2024-08-14
> > > >
> > > > Dear Reviewer gv1w,
> > > >
> > > > We sincerely appreciate your insightful feedback and thoughtful suggestions, which have been instrumental in improving the quality of our paper. We will throughly discuss the aforementioned works，including but not limited to DynamicViT, EViT, ToMe, DiffRate, Conditional Adapters, AdaMix, and MoLoRA in the revision.
> > > >
> > > > Our approach, DyT, contributes a novel exploration of data-dependent computation in vision transformer using a PEFT manner. While our current work focuses on the vision transformer due to computational resource limitations,  we believe this work would be more valuable in this era, when combined with large language models or multi-modal models.
> > > >
> > > >
> > > > We are more than willing to address any additional questions you may have. Once again, thank you for your support.
> > > >
> > > > Best Regards
> > > >
> > > > Authors

---

### Official Review · Reviewer_Zpyf · 2024-07-10

**Soundness:** 3
**Presentation:** 4
**Contribution:** 3
**Rating:** 7
**Confidence:** 4

**Summary:**

This paper proposes Dynamic Tuning (DyT) to improve both parameter and inference efficiency for ViT adaptation. DyT inserts a token dispatcher for each transformer block to learn to dynamically determine whether a token should be activated or deactivated.

**Strengths:**

- This paper focuses on **inference efficiency**, which is an important problem for adapting large pre-trained models to downstream tasks using PEFT methods.
- This paper proposes Dynamic Tuning to improve the inference efficiency for ViT adaptation. Extensive experiments and analyses validate the proposed method.

**Weaknesses:**

I don't see any major weakness in the paper, but I have a minor concern about its novelty. The core technique in Dynamic Tuning seems to be **token pruning** without explicitly reducing the number of tokens, but token pruning itself is not a novel technique. This paper does not well demonstrate the difference between existing dynamic token pruning methods and the proposed method.

**Questions:**

See Weakness

**Limitations:**

Yes, the authors discuss limitations in the Discussion and Conclusion section.

---

> ### Author Rebuttal · Authors · 2024-08-06
>
> > #### W1: The core technique in Dynamic Tuning seems to be token pruning without explicitly reducing the number of tokens. This paper does not well demonstrate the difference between existing dynamic token pruning methods and the proposed method.
>
> Thanks. We would like to kindly clarify that our method is **not a token pruning** method but rather **a token skipping method**. In our approach, some tokens skip the original blocks and only pass through the adapter, **preserving the feature map** completely. Below, we list the differences between our method and representative token pruning methods (DynamicViT[1], EViT[2], ToMe[3]) below. We will include these in Section 2 in the revision.
>
>
> | Method     | Training     |    Efficient Strategy  | FLOPs for different samples | Classification | Dense Prediction | Target dataset
> | -------- | -------- | -------- | -------- | -------- |-------- |--------
> | DynamicViT | Full-parameter |  Learn to hierarchically keep P% tokens at certain layers (e.g. 3th,6th 9th layers in ViT-B) | same | ✅ | ❌ | In-domain
> | EViT |  Full-parameter |   Only top-K attentive tokens and fuse the inattentive tokens at certain layers (e.g. 4th,7th, and 10th layer in ViT-B) | same | ✅  | ❌ | In-domain
> | ToMe |  Training free | Prune through merging N tokens via similarity in each layer |  same | ✅| ❌ Need modification | In-domain
> | DyT (Ours) | Parameter-efficient | Learn how many tokens to skip before which MLP block | data-dependent | ✅ | ✅ |  Cross-domain
>
> Note that, our method can be seamlessly combined with ToMe (Table 5)
>
> **Analysis:**
> - Compared to these methods, our approach does **not rely on a manually set number or percentage** of tokens to prune. Instead, it **learns where and how many tokens to skip**, constrained by the average activation rate loss $L_{rate}$.
> - Our model allocates computation in a **data-dependent** manner, which varies between different samples.
> - Our method can also handle the dense prediction task, whereas  other methods fail due to the incomplete feature map.
> - Previous token pruning methods primarily focues on accelerating the model within the same dataset used for pretraining, while DyT aims to improve efficiency during **cross-domain adaption**.
>
>
> > ### Reference
>
> [1] Dynamicvit: Efficient vision transformers with dynamic token sparsification, 2021
>
> [2] Expediting Vision Transformers via Token Reorganizations, 2022
>
> [3] Token Merging: Your ViT But Faster, 2022

---

> ### Author Response · Authors · 2024-08-12
>
> Dear Reviewer Zpyf,
>
> Thanks so much again for the time and effort in our work. Considering the limited time available and to save the reviewer's time, we summarize our responses here.
>
> 1. **[Difference between existing dynamic token pruning methods and the proposed method]**
>
> **Response:**
>
> - Our model learns where and how many tokens to skip.
>
> - Our model allocates computation in a data-dependent manner.
>
> - Our model preserves complete feature maps and can handle denste prediction tasks.
>
> - Our model aims to improve efficiency during cross-domain adaption.
>
>
> Since the discussion stage is already halfway through, may we know if our rebuttal addresses the concerns?
> If there are further concerns or questions, we are more than happy to address them. Thanks again for taking the time to review our work and provide insightful comments.
>
>
> Best Regards
>
> Authors

---

### Official Review · Reviewer_uJS2 · 2024-07-11

**Soundness:** 3
**Presentation:** 3
**Contribution:** 3
**Rating:** 6
**Confidence:** 5

**Summary:**

The paper introduces a method called Dynamic Tuning (DyT) designed to enhance both parameter and inference efficiency when adapting Vision Transformers (ViTs) for various visual tasks. DyT employs a token dispatcher to selectively process only the most informative tokens, reducing computational redundancy. Additionally, an enhanced adapter inspired by the mixture-of-experts mechanism is introduced to boost performance with minimal extra cost. The method is comprehensively evaluated across tasks like image and video recognition and semantic segmentation, demonstrating superior performance and significantly reduced FLOPs compared to existing parameter-efficient fine-tuning methods. The results underscore DyT's effectiveness in achieving state-of-the-art performance while being computationally efficient.

**Strengths:**

1. The paper is well written and easy to follow, the figures are clear and easy to understand.
2. Most previous PEFT methods often focus on decreasing tunable parameters in training to conserve memory. However, these approaches do not lessen computational load during inference (and may even increase it, as seen in Adapter, Prompt, etc.). This paper introduce token pruning to help to reduce model inference costs.

**Weaknesses:**

1. The authors introduced additional loss functions for DyT, does this affect the speed of convergence?

1. Gumbel-Sigmoid is a prevalent technique for maintaining end-to-end training. However, its computational outcomes may vary compared to Sigmoid. It is valuable to explore the disparities in the calculated results of these two methods.

1. Certain experimental phenomena may become more transparent with additional explanations provided within the text. For instance, **(a)** Image Accuracy in Table 2 demonstrates optimal performance when excluding MoE, and it appears that the performance does not exhibit an increase proportional to the number of experts. **(b)** Meanwhile, for N=12, the FLOPs are paradoxically lower than when N=8. **(c)** In Table 6, the FLOPs of DyT N = 4 is higher than Dynamic-Full, this seems counterintuitive. **(d)** In Table 10, the decrease in FLOPs for DyT is not as significant as in VTAB-1K, and the FLOPs for DyT N=4 is higher than the version not adapted to the MoE. *The reasons behind these observations warrant further exploration*.

**Questions:**

1. I am concerned about the potential performance impact of token pruning on the detection task. Have the authors conducted any experiments in this regard?

2. I came across a paper [1] that dynamically assesses token significance, and it would be beneficial to discuss and cite it in the article.




---
   [1] Dynamic Adapter Meets Prompt Tuning: Parameter-Efficient Transfer Learning for Point Cloud Analysis[C]//CVPR. 2024.

**Limitations:**

The authors have discussed the limitations.

---

> ### Author Rebuttal · Authors · 2024-08-06
>
> > #### W1: Does introduced additional loss functions affect the speed of convergence
>
> Thanks. The introduced additional loss functions do not impact the speed of convergence. We plot the loss values throughout the fine-tuning process and record the test accuracy at every training epoch. Please find the figure in the **attachment PDF**.
> The explanation and figures here will be included in the Appendix, with a new section discussing the convergence speed.
>
> > #### W2: It is valuable to explore the disparities in the calculated results of Sigmoid and Gumbel-Sigmoid.
>
> Thanks. We would like to clarify that DyT **only employs the Gumbel-Sigmoid during training** and replaces it with Sigmoid in the inference. Based on the suggestion, we conduct an experiment using Sigmoid during both training and inference, as shown in the table below:
>
> **Setting:**
> "DyT-" denotes using Sigmoid during both training and inference. We report the average FLOPs on CIFAR-100.
> |Method|FLOPS|CIFAR-100|SVHN|Food-101
> |-|-|-|-|-
> |DyT|12.21|91.37|97.08|90.32
> |DyT-|12.52|88.48|96.63|85.41
>
> **Analysis & Conclusion**
> - Replacing the Gumbel-Sigmoid with Sigmoid negatively impacts performance, as it results in a non-differentiable problem and increases the difficulty of fine-tuning (Lines 137-149). We will include this experiment in the Appendix in the revision.
>
>
> > #### W3: Certain experimental phenomena may become more transparent with additional explanations
>
> Thanks. These valuable questions and explanations will be added to the “Frequent Questions” section in the Appendix of the revised manuscript.
>
> |Question|Answer|
> |-|-|
> |(a) Image Accuracy in Table 2 demonstrates optimal performance when excluding MoE | The aim of the MoE-adapter is to **tackle challenging tasks** e.g. video classification and dense prediction, without introducing extra computational cost. For relatively easy image tasks, introducing more tunable parameters through MoE does not bring benefits.
> |It appears that the performance does not exhibit an increase proportional to the number of experts | Performance improves from $N=2$ to $N=4$, achieving the best results on video tasks. This indicates that **an appropriate number** of experts can enhance performance. However, too many experts ($N=8$ and $N=12$) increase the difficulty of fine-tuning and degrade performance.
> |(b) Meanwhile, for N=12, the FLOPs are paradoxically lower than when N=8 | Theoretically, the MoE-adapter with any number of experts should have similar FLOPs to the ordinary adapter (Line 202-204). Meanwhile, the actual activation rate during inference depends on the **learned token dispatcher** after fine-tuning, resulting in **slight fluctuations in FLOPs** between different models. These exlain why DyT N=12 may have lower FLOPs than DyT N=8 |
> |(c) In Table 6, the FLOPs of DyT N=4 is higher than Dynamic-Full, this seems counterintuitive.| First, "Dynamic-Full" does not have additional cost of adapters compared to DyT and DyT $N=4$. Additionally, the **learned token dispatcher** varies between models, leading to slight differences in actual token activation rates during inference. This results in the posibility of Dynamic-Full (12.24 GFLOPs) being slightly more efficient than DyT $N=4$ (12.29 GFLOPs).
> | (d) In Table 10, the decrease in FLOPs for DyT is not as significant as in VTAB-1K | On one hand, in all experiments, DyT is applied **only to the Vision Transformer**. The vision transformer in Table 10 equips a segmentation head from UperNet [4] for semantic segmentation. On the other hand, the proposed "MLP Dispatch" in DyT does not reduce the FLOPs in self-attention, whose FLOPs increase with the increasing size of the feature map. This reuslts in a less significant FLOPs reduction ratio compared to VTAB-1K.
> | and the FLOPs for DyT N=4 is higher than the version not adapted to the MoE. |  In Table 10, the FLOPs of DyT $N=4$ (584.40) is slightly lower than DyT (584.57).  This slight difference is introduced by the **learned token dispatcher** when the fine-tuning convergence.
>
> > #### Q1: Potential performance impact of token pruning on the detection task.
>
> Thanks for this valuable suggestion. We conduct experiments on the COCO. Unlike token pruning, DyT employs a **token skipping** method, enabling it to effectively address the object detection task.
>
> **Setting:**
> we adopt ViTDet[2] as the detector. We fine-tune the model for 12 epochs on the COCO dataset. The bottleneck dimension $d$ in adapters is set to 128. Here, we only measure the **FLOPs in the vision transformer and feature pyramid**.
> |Method|FLOPs(G)|Tunable Param.(M) in backbone|BBox AP|Seg AP
> |-|-|-|-|-
> |Full tuning|554.86|85.80|44.57|39.78
> |AdaptFormer|564.531|2.56|38.71|35.27
> |DyT|468.40|2.56|39.78|36.11
> |DyT $N=4$|466.32|10.24|40.97|37.11
>
> **Analysis & Conclusion:**
> - The full-tuning method performs the best, likely due to the significant gap between bounding box regression and the pretraining of the vision transformer, requring more parameters in the backbone to be updated.
> - DyT demonstrates superior performance compared to the AdapterFormer[3] with fewer FLOPs, validating the effectiveness of our approach.
> - DyT with the proposed MoE-adapter (DyT $N=4$) surpasses the original DyT, highlighting the importance of the MoE-adapter.
>
> > #### Q2: Paper [1] dynamically assesses token significance, and it would be beneficial to discuss and cite it
>
> We appreciate this valuable suggestion. We carefully read the paper and will cite it by "Zhou et.al [1] propose assessing token significance with a ReLU function to effectively tackle point cloud analysis" in Related Work.
>
> > #### Reference
>
> [1] Dynamic Adapter Meets Prompt Tuning: Parameter-Efficient Transfer Learning for Point Cloud Analysis, 2024
>
> [2] Exploring plain vision transformer backbones for object detection, 2022
>
> [3] Adaptformer: Adapting vision transformers for scalable visual recognition, 2022
>
> [4] Unified perceptual parsing for scene understanding, 2018

---

> > ### Comment · Reviewer_uJS2 · 2024-08-12
> >
> > I carefully read the author's rebuttal and the comments from other reviewers. The rebuttal addressed most of my concerns. While it’s evident that DyT shares similarities with token pruning methods, its dynamic token selection strategy for PEFT is valid. Most existing PEFT methods focus solely on reducing training costs, often overlooking inference overhead.
> >
> > I also notice a performance gap between PEFT and FFT in the object detection task in the author's response. I encourage the authors to include relevant tables and analysis in the next version to guide future work. Additionally, the differences with other studies highlighted by reviewers could be addressed in the next version's appendix.
> >
> > Based on these considerations, I am inclined to accept this paper.

---

> ### Author Response · Authors · 2024-08-12
>
> Dear Reviewer uJS2,
>
> Thanks so much for the support. We sincerely appreciate the valuable feedback.
>
> We will include all experimental results and analysis in the revision to facilitate future work. We will clarify the differences with other studies highlighted by reviewers in the revised paper.
>
> We hope our response resolves your concerns. We are more than willing to answer any further questions you may have. Your support is appreciated and helps us improve our work.
>
> Best regards
>
> Authors

---

### Official Review · Reviewer_fmWb · 2024-07-13

**Soundness:** 4
**Presentation:** 3
**Contribution:** 3
**Rating:** 6
**Confidence:** 4

**Summary:**

This paper proposes Dynamic Tuning, an efficient tuning method to enhance both parameter and inference efficiency for ViT adaptation. The method improves efficiency by dropping some tokens entering pre-trained layers through a token dispatcher while forwarding all tokens to the adapter, maintaining performance. Additionally, the method leverages a Mixture of Experts (MoE) adapter to enhance adaptation performance on certain datasets. The proposed approach achieves comparable or superior performance with significantly fewer FLOPs compared to traditional parameter-efficient transfer learning methods on the VTAB-1K dataset.

**Strengths:**

- The proposed method is the first to enhance inference efficiency and use fewer parameters for fine-tuning by drawing inspiration from dynamic neural networks, achieving higher efficiency and performance than existing methods.

- The paper is well-written and easy to understand, with a clear and intuitive explanation of the proposed method.

- The experiments are extensive, demonstrating results on various datasets and providing detailed analyses, such as model variants, the number of MoE experts, and the layers where token dropping occurs. This thorough experimentation helps in-depth understanding of the proposed method.

**Weaknesses:**

- The application of MoE in the paper detracts from the overall understanding and value of the proposed method. MoE improves performance only in video classification, not image classification, and is absent from most experiments. Furthermore, Table 6 shows that using MoE results in fewer FLOPs, which is theoretically implausible. The paper lacks an analysis explaining this and only provides experimental results, which hinders comprehension. MoE's inclusion in the main paper is questionable and might be better suited for the appendix.

- The proposed method may appear as a simple combination of dynamic neural networks and adapters. While it differs in learning token dropping during fine-tuning, it essentially drops tokens from the original block while using all tokens in the adapter. The token dispatch mechanism is similar to existing dynamic neural network ideas. The paper should report the performance of DynamicViT and EViT's full fine-tuning and their combination with other PEFT methods in Table 5. If the proposed method performs better, an analysis explaining this phenomenon is necessary.

- There are several instances where reported FLOPs differ significantly from theoretical values, requiring more detailed explanations. For instance, in Table 1, it is unclear why MLP Dispatch has lower FLOPs than Layer Dispatch, given that the attention layer uses all tokens in MLP Dispatch but fewer tokens in Layer Dispatch.

- While the proposed method achieves higher performance despite added constraints compared to existing PEFT methods, the paper lacks explanations or analyses for this improvement. The authors should explain why their method outperforms traditional adapters.

**Questions:**

- The authors should consider removing the MoE part from the paper, as it does not significantly contribute to the overall method.

- Discuss the increased training time caused by the need for two forwards (with and without the complete model) during training as a limitation.

- The paper should include an ablation study on the loss function. Given the significant performance differences observed in the distillation ablation study in the appendix, a comprehensive description of the loss function's impact is necessary.

**Limitations:**

The authors have adequately addressed some limitations of their work. However, they did not discuss the increased training time as a limitation. There are no negative societal impacts.

---

> ### Author Rebuttal · Authors · 2024-08-06
>
> We will include all explanations and results in the revision.
>
> > #### W1:MoE improves performance only in video classification, not image classification, and is absent from most experiments
>
> We appreciate this comment. We would like to clarify that the MoE-adapter is primarily designed to enhance the adapter’s capacity to **address challenging tasks** (Lines 180-183). In addition to video classification, we validate its effectiveness in semantic segmentation (Appendix Table 10) and object detection (In reponse to Reviewer uJS2). We do not employ it in the VTAB-1K experiments as it does not outperform the model without it, likely due to the extremely limited training data.
>
> > #### W2:Table 6 shows that using MoE results in fewer FLOPs, which is theoretically implausible
>
> Thanks. We would like to clarify that it is reasonable for the model with the MoE adapter to result in fewer FLOPs:
> - The FLOPs of our models depend on **learned token dispather** during fine-tuning and may slightly **fluctuate around the target FLOPs** (controlled by $L_{rate}$).
> - The extra computational cost of the adapter and the MoE adapter is nearly equivalent (Lines 202-204).
>
> > #### W3 & Q1:MoE might be better suited for the appendix
>
> We appreciate this valuable suggestion. In the revision, we will move the introduction of the MoE adapter and its corresponding experiments to the Appendix.
>
> > #### W4:The token dispatch mechanism is similar to existing dynamic neural network
>
> Thanks. We present the differences below:
> |Method|Training|Efficient strategy|FLOPs for different samples| Classification|Dense Prediction|Target dataset
> |-|-|-|-|-|-|-
> |DynamicViT|Full-parameter|Learn to keep P% tokens at certain layers (e.g. 3th,6th 9th layers in ViT-B)|same|✅|❌|In-domain
> |EViT|Full-parameter|Only keep top-K attentive tokens and fuse other tokens at certain layers (e.g. 4th,7th, and 10th layer in ViT-B)|same|✅|❌|In-domain
> |DyT (Ours)|Parameter-efficient|Learn how many tokens to skip before which MLPs|data-dependent|✅|✅|Cross-domain
>
> **Analysis & Conclusion:**
> - Dynamic neural networks typically prune a **certain number or percentage** of tokens in fixed layers, while our token dispatcher **learns the number of tokens to skip** before each MLP block.
> - DynamicViT and EViT maintain the **same FLOPs for all samples**, while DyT achieves **data-dependent computation**, which is more flexible and reasonable.
> - Instead of token pruning, the token skipping in our token dispatcher allows the DyT to preserve **complete feature maps**, enabling it to perform dense prediction tasks.
>
> > #### W5:The paper should report the performance of DynamicViT and EViT's full fine-tuning and their combination with other PEFT methods
>
> Sincerely thanks. We present the results below.
>
> **Setting:**
> We combine DynamicViT and EViT with AdaptFormer. The bottleneck dimmension is set to 8.
> |Method|VTAB-1K accuracy|FLOPs(G)|Tunable Param.(M)| Throughput (img/s)
> |-|-|-|-|-
> DynamicViT|60.10|14.05|88.70|1010.40
> DynamicViT+AdaptFormer|75.48|14.23|3.10|954.82
> EViT|67.42|11.62|85.8|1233.45
> EViT+AdaptFormer|74.63|11.77|0.16|1152.38
> DyT r=0.5 |77.17|12.54|0.16|912.39
> DyT r=0.5 + ToMe|76.60|9.85|0.16|1114.70
>
> **Analysis and Conclusion:**
> - Combining DynamicViT and EViT with AdaptFormer enhances performance, validating the significance of exploring both parameter and inference efficiency for vision transformers
> - DyT outperforms “DynamicViT+AdaptFormer” and “EViT+AdaptFormer” by learning to skip an appropriate number of tokens at each MLP block, rather than pruning a fixed number (EViT) or percentage (DynamicViT) of tokens at specific layers. Figure 5 in the main paper illustrates that different datasets benefit from varying token-skipping strategies.
>
> > #### W6:It is unclear why MLP Dispatch has lower FLOPs than Layer Dispatch
>
> Thanks. There are two reasons:
> - The actual token activation rate during inference depends on the **learned token dispatcher,** causing real FLOPs to fluctuate around the theoretical value.
> - We do not strictly control FLOPs across the four variants. Specifically, we set the activation rate $r$ to 0.5 for “Attention Dispatch” and “MLP Dispatch” variants, and to 0.7 for “Attention-MLP Dispatch” and “Layer Dispatch” variants, ensuring **similar** average FLOPs. Figure 6 (Appendix) shows that “MLP Dispatch” consistently achieves the best performance under similar FLOPs.
>
> > #### W7:The authors should explain why their method outperforms traditional adapters
>
> Thanks. We list the explanations below:
> - The dynamic architecture in DyT enhances generalization. It introduces a form of disturbance in the input data, akin to Dropout. This mechanism is particularly crucial when training data is limited e.g. VTAB-1K.
> - The distillation loss in DyT. We adopt the complete model as the teacher of the dynamic model, significantly enhancing performance. Such a self-distillation mechanism is only available in the dynamic architecture.
> - Previous work [1] and DynamicViT also show dynamic architectures outperforming static models with fewer FLOPs.
>
> > #### Q2&L1:Discuss the increased training time caused by the need for two forwards
>
> Thanks. We acknowledge that training with two forward passes takes 1.8 times longer than only one pass, but this significantly enhances performance without compromising parameter and inference efficiency. Given that our primary contribution focuses on improving inference efficiency, the additional training time would be acceptable.
>
> > #### Q3:The paper should include an ablation study on the loss function
>
> Thanks. We present the results below.
> |Method|VTAB-1K accuracy
> |-|-
> |DyT|77.14
> |DyT w/o $L_{distill}$|76.70|
> |DyT w/o $L_{distill}$ & $L^\prime_{cls}$| 75.73
>
> **Analysis and Conclusion:**
>
> Removing $L_{distill}$ or $L^\prime_{cls}$ negatively impacts performance, validating the effectiveness of loss functions in DyT.
>
> > #### Reference:
>
> [1] Latency-aware Unified Dynamic Networks for
> Efficient Image Recognition. 2024.

---

> ### Author Response · Authors · 2024-08-12
>
> Dear Reviewer fmWb,
>
>
> Thanks so much again for the time and effort in our work. Considering the limited time available and to save the reviewer's time, we summarize our responses here.
>
>
> > ### 1. [MoE improves performance only in video classification, not image classification, and is absent from most experiments]
>
> **Response:**
> - MoE-adapter is primarily designed to enhance the adapter’s capacity to address challenging tasks (Lines 180-183).
> - We also validate its effectiveness in semantic segmentation (Appendix Table 10) and object detection (In reponse to Reviewer uJS2).
> - We do not employ it in the VTAB-1K experiments as it does not outperform the model without it, likely due to the extremely limited training data
>
>
> > ### 2. [Table 6 shows that using MoE results in fewer FLOPs, which is theoretically implausible]
>
>
> **Response:**
> - The FLOPs of our models depend on **learned token dispather** during fine-tuning and may slightly **fluctuate around the target FLOPs**.
> - The extra computational cost of the adapter and the MoE adapter is nearly equivalent (Lines 202-204).
>
>
> > ### 3. [MoE might be better suited for the appendix]
>
> **Response:**
> - In the revision, we will move the introduction of the MoE adapter and its corresponding experiments to the Appendix.
>
>
> > ### 4.[The token dispatch mechanism is similar to existing dynamic neural network]
>
>
> **Response:**
> - DynamicViT and EViT prune a **certain number** or percentage of tokens in fixed layers, while our token dispatcher **learns the number of tokens to skip** before each MLP block.
>
> - DynamicViT and EViT maintain the **same FLOPs for all samples**, while DyT achieves **data-dependent computation**.
>
> - DyT to preserve **complete feature maps**
>
>
> > ### 5.[The paper should report the performance of DynamicViT and EViT's full fine-tuning and their combination with other PEFT methods]
>
> **Response:**
> - Combining DynamicViT and EViT with AdaptFormer enhances performance.
> - DyT outperforms “DynamicViT+AdaptFormer” and “EViT+AdaptFormer” by learning to skip an appropriate number of tokens at each MLP block.
>
>
> > ### 6.[It is unclear why MLP Dispatch has lower FLOPs than Layer Dispatch]
>
> **Response:**
> - The actual token activation rate during inference depends on the **learned token dispatcher**.
> -  We set the activation rate $r$ to 0.5 for “Attention Dispatch” and “MLP Dispatch” variants, and to 0.7 for “Attention-MLP Dispatch” and “Layer Dispatch” variants, ensuring **similar** average FLOPs.
>
>
>
> > ### 7.[The authors should explain why their method outperforms traditional adapters]
>
> **Response:**
>
> - The dynamic architecture in DyT enhances generalization.
> - The distillation loss in DyT improves performance.
> - Dynamic architectures outperforming static models is also observed in previous works.
>
>
>
> > ### 8.[Discuss the increased training time caused by the need for two forwards]
>
> **Response:**
> - Two forward passes takes 1.8 times longer than only one pass
> - Given that our primary contribution focuses on improving inference efficiency, the additional training time would be acceptable.
>
>
>
> > ### 9.[The paper should include an ablation study on the loss function]
>
>
> **Response:**
>
> - Removing $L_{distill}$ or $L^\prime_{cls}$ negatively impacts performance.
>
>
> Since the discussion stage is already halfway through, may we know if our rebuttal addresses the concerns?
> If there are further concerns or questions, we are more than happy to address them. Thanks again for taking the time to review our work and provide insightful comments.
>
>
> Best Regards
>
>
> Authors

---

> > ### Comment · Reviewer_fmWb · 2024-08-14
> >
> > Thank you for your clear and concise response. I have reviewed all the reviews and rebuttals, and I am pleased to say that all of my concerns have been fully addressed. While the token pruning and PEFT involing the proposed method are not novel concepts on their own, I acknowledge that the combination of these techniques in the proposed approach is meaningful and has led to interesting and significant experimental results. As such, I will be raising my score.
> >
> > However, there is one point of concern I would like to highlight. While data-dependent FLOPs can be an advantage in certain scenarios, they may often pose a drawback. In many cases, it is more practical for FLOPs to be adjusted based on available computational resources rather than varying with the data, especially when working within specific resource constraints.

---

> > > ### Author Response · Authors · 2024-08-14
> > >
> > > Dear Reviewer fmWb,
> > >
> > > We sincerely appreciate your support and suggestions! We will include all experimental results and follow the suggestions in the revision to improve our work.
> > >
> > > Our model, DyT, contributes a novel exploration of the data-dependent computation in vision transformer through a PEFT manner. As you have kindly pointed, it primarily focuses on improving the inference efficiency when the device has enough capacity to run the model. Adapting it to different resource scenarios requires training DyT with different FLOPs targets. We believe the comment of a resources-dependent model is quite insightful and promising. This valuable suggestion inspires us to explore a design that has both the advantages of data-dependent and resource-dependent models.
> > >
> > > We are more than willing to answer any further questions you may have. Your support is appreciated and helps us improve our work. Thanks again.
> > >
> > > Best Regards
> > >
> > > Author

---

### Author Rebuttal · Authors · 2024-08-06

Dear ACs and reviewers,

We sincerely appreciate the time and effort provided by all reviewers and ACs in our work. In particular, we are encouraged to see that Reviewer fmWb finds that our method **"achieves higher efficiency and performance than existing methods"**. Reviewer uJS2 thinks the proposed method **”is comprehensively evaluated across tasks“** and demonstrates **"superior performance and significantly reduced FLOP“**. Reviewer Zpyf **“don't see any major weakness in the paper"**. Reviewer gv1w acknowledges that the problem proposed in this paper **"is very meaningful"**.

We will continue to refine our work based on the feedback.


Thanks,

Authors of submission 3553

---

### Decision · Program_Chairs · 2024-09-25

**Decision:**

Accept (poster)

**Comment:**

The reviewers recognized that the proposed approach is novel and achieves both efficiency and performance. They also appreciated the clarity of the manuscript and extensive experiments. However, at the same time, the reviewers raised concerns with incremental novelty, lack of detailed discussion on experimental results, convergence speed, and some presentation issues. These concerns are successfully addressed by the rebuttal and subsequent responses in the discussion. Consequently, the reviewers unanimously championed the paper after the discussion period. The AC agrees with the reviewers and recommends acceptance. The authors are strongly encouraged to polish the manuscript according to the valuable comments and to add new results and discussions brought up in the responses to the revision.